# MIXING CORRUPTED PREFERENCES FOR ROBUST AND FEEDBACK-EFFICIENT PREFERENCE-BASED REINFORCEMENT LEARNING

## ABSTRACT

Preference-based reinforcement learning (RL) trains agents using non-expert feedback without the need for detailed reward design. In this approach, a human teacher provides feedback to the agent by comparing two behavior trajectories and labeling the preference. Although recent studies have improved feedback efficiency through methods like unsupervised exploration to collect various trajectories and combined self- or semi-supervised learning for unlabeled queries, they often assume flawless human annotation. In practice, human teachers might make mistakes or have conflicting opinions about trajectory preferences. The potential negative impact of such corrupted preferences on capturing user intent remains an underexplored challenge. To address this challenge, we introduce mixing corrupted preferences (MCP) for robust and feedback-efficient preference-based RL. Mixup has shown robustness against corrupted labels by reducing the influence of faulty instances. By generating new preference data through the component-wise mixing of two labeled preferences, our method lessens the impact of corrupted feedback, thereby enhancing robustness. Furthermore, MCP improves feedback efficiency: even with limited labeled feedback, it can generate unlimited new data. We evaluate our method on three locomotion and six robotic manipulation tasks in B-Pref benchmark, comparing it with PEBBLE in contexts with both perfectly rational and imperfect teachers. Our results show that MCP significantly outperforms PEBBLE, requiring fewer feedback instances and a shorter training period, highlighting its superior feedback efficiency.

## 1 INTRODUCTION

Deep reinforcement learning (DRL) has shown remarkable progress in addressing sequential decision-making tasks across various domains (Kober et al., 2013; Mnih et al., 2015; 2016; Schulman et al., 2017; Haarnoja et al., 2018a;b). In DRL, configuring an appropriate reward function is important for the agent to successfully solve the task at hand. For instance, in robotic manipulation tasks, a reward function may need to be designed to encourage a robot to move forward without falling down or to successfully grasp an object and relocate it accurately. However, crafting an appropriate scalar reward can be challenging because of the complexity of tasks (Christiano et al., 2017). Such detailed reward design often requires a deep domain knowledge and a comprehensive understanding of the environment, which can pose a significant obstacle when applying DRL to complex real-world problems broadly (Yu et al., 2019a).

Preference-based reinforcement learning (RL) offers an alternative method for training agents without the need for domain-specific reward design (Christiano et al., 2017; Lee et al., 2021b; Park et al., 2022; Liang et al., 2022). Unlike traditional RL, which relies on explicitly defined scalar rewards, preference-based RL exploits human feedback in the form of preferences. Each preference consists of two behavior trajectories along with a binary label, indicating the superior one. This relative comparison makes it easier to assess agent performance without specific reward values. In preference-based RL, the reward estimator plays an important role as an additional approximator with learnable parameters. It determines rewards for each transition from human feedback, aligning agent behaviors with the preferred actions from the feedback. Consequently, preference-based RL

addresses the challenge of reward specification in complex tasks and enhances learning in real-world problems (Hejna III & Sadigh, 2023; Jain et al., 2023).

Existing studies on preference-based RL typically assume that human teachers can provide perfect preferences. However, in practice, non-expert human teachers might occasionally reverse their preferences, have difficulty determining the better choice, or even give judgments that are inconsistent with others (Lee et al., 2021a). While these inconsistent and corrupted preferences can significantly affect the performance of preference-based RL methods, solutions to mitigate the challenges caused by such corrupted preferences have not yet been identified.

Various methods have been studied to address the challenge of corrupted labels (Zhang et al., 2018; Han et al., 2018; Jiang et al., 2018; Arazo et al., 2019; Jiang et al., 2020; Li et al., 2020). Among them, mixup (Zhang et al., 2018) stands out for its remarkable robustness against label corruption. This is achieved by reducing the memorization of specific instances using vicinal risk minimization (VRM) principle (Chapelle et al., 2000). One key advantage of mixup is that it does not require intricate methods to detect corrupted data. Additionally, mixup improves data efficiency and enhances generalization by generating new instances even when with a limited dataset.

In this study, our objective is to implement mixup in the reward estimator learning procedure of preference-based RL to enhance robustness against corrupted preferences and improve feedback efficiency. To this end, we propose mixing corrupted preferences (MCP). We outline the process of mixing two feedback instances. Each instance consists of a pair of two trajectory segments (i.e., state-action sequences) and a corresponding preference label. For a pair of two feedback instances, we mix the first trajectory segments from the pair, and similarly, we do so for the second segments. The mixed trajectory segment of each index is convex combination of two trajectory segments with a specific weight. Additionally, the preference label of the new trajectory segments pair is also determined as the convex combination of the corresponding labels, again using the same weight.

The newly generated preference data through MCP alleviates the adverse effects of accidentally corrupted preferences because MCP reduces the memorization and improves generalization, similar to the original mixup. In addition, MCP improves feedback efficiency even when preference data is scarce by enlarging the preference dataset through mixup without limitation. We evaluate our method on six robotic manipulation tasks and three locomotion tasks on the B-Pref benchmark (Lee et al., 2021a), which provides diverse mechanisms of corrupting preferences. Our experiments demonstrate that MCP significantly outperforms previous preference-based RL methods in managing corrupted preferences. Furthermore, we show high feedback efficiency of MCP, rapidly approaching the performance levels of soft actor-critic (SAC) (Haarnoja et al., 2018b) with true rewards, even when working with minimal feedback.

## 2 PRELIMINARIES

### 2.1 FITTING THE REWARD FUNCTION WITH HUMAN PREFERENCES

In the preference-based RL framework, we assume that the specific rewards from the environment do not exist. Instead, we assume that there is a human teacher providing the preference between two behavior trajectories of the agent to teach the reward estimator $\hat{r}_\psi$, as in previous works (Christiano et al., 2017; Lee et al., 2021b). Concurrently, we train agent with the estimated reward derived from $\hat{r}_\psi$ . Formally we denote a trajectory segment $\sigma$ as a state-action sequence with fixed length $H$, $\sigma^l = \{(s_1^l, a_1^l), (s_2^l, a_2^l), ..., (s_H^l, a_H^l)\} \quad \forall l \in (0, 1)$. Given a pair of trajectory segments $(\sigma^0, \sigma^1)$, human teachers indicate their preference $y$, which is a label distribution consisting of two categories. The probability mass is one if a segment is selected and zero otherwise. If the human teacher cannot judge one's superiority, then each mass of two choices is assigned 0.5. That is, $y$ can be one of (1,0), (0,1), and (0.5,0.5). The entire instance $(\sigma^0, \sigma^1, y)$ is stored in preference dataset $D = \{(\sigma^0, \sigma^1, y)_i\}_{i=1}^N$, where $N$ represents size of the dataset. Then, according to the Bradley-Terry model (Bradley & Terry, 1952), we can model a preference predictor with the reward estimator $\hat{r}_\psi$ as follows:

$$P_\psi(\sigma^1 \succ \sigma^0) = \frac{\exp\left(\sum_{t=1}^H \hat{r}_\psi(s_t^1, a_t^1)\right)}{\exp\left(\sum_{t=1}^H \hat{r}_\psi(s_t^1, a_t^1)\right) + \exp\left(\sum_{t=1}^H \hat{r}_\psi(s_t^0, a_t^0)\right)}, \tag{1}$$

where $(\sigma^1 \succ \sigma^0)$ indicates that trajectory segment $\sigma^1$ is more preferable to trajectory segment $\sigma^0$. Given the preference dataset $D$, the reward estimator can be trained by minimizing the binary cross entropy (BCE) loss between ground truth preferences and predicted preferences:

$$
\begin{aligned}
L_\psi &= E_{(\sigma^0,\sigma^1,y)\sim D}[L^{Reward}(\sigma^0,\sigma^1,y)] \\
&= -E_{((\sigma^0,\sigma^1),y)\sim D)}[y(0)\log P_\psi(\sigma^0 \succ \sigma^1) + y(1)\log P_\psi(\sigma^1 \succ \sigma^0)],
\end{aligned}
\tag{2}
$$

where $y(i)$ indicates the $i$th component of $y$.

### 2.2 Feedback provision using simulated human teacher

Instead of providing direct rewards, we used the simulated human teacher from B-Pref benchmark (Lee et al., 2021a) for feedback provision. This teacher is defined as:

$$
\begin{aligned}
&P(\sigma^1 \succ \sigma^0; \eta_{stoc}; \eta_{myopic}) \\
&= \frac{\exp\left(\eta_{stoc}\sum_{t=1}^H \eta_{myopic}^{H-t} r(s_t^1, a_t^1)\right)}{\exp\left(\eta_{stoc}\sum_{t=1}^H \eta_{myopic}^{H-t} r(s_t^1, a_t^1)\right) + \exp\left(\eta_{stoc}\sum_{t=1}^H \eta_{myopic}^{H-t} r(s_t^0, a_t^0)\right)},
\end{aligned}
\tag{3}
$$

where $r(s,a)$ denotes the ground truth reward for a state-action transition. $\eta_{stoc} \in [0,\infty)$ is a rationality constant and $\eta_{myopic} \in (0,1]$ is a myopia discount factor. A higher $\eta_{stoc}$ makes the teacher more rational and deterministic. If $\eta_{stoc} = 0$, the decisions are random. A lower $\eta_{myopic}$ gives more weights to earlier transitions compared to later ones.

Three additional hyperparameters are introduced to simulate different forms of human irrationality: $\epsilon, \delta_{equal}$, and $\delta_{skip}$. With a probability $\epsilon$, the teacher mistakenly flips a preference label. If the cumulative rewards for two choices are too close to distinguish, denoted as $\left|\sum_{t=1}^H r(s_t^1, a_t^1) - \sum_{t=1}^H r(s_t^0, a_t^0)\right| < \delta_{equal}$, they are treated as equally preferable, each getting a weight of 0.5 in the preference distribution. When neither segment meets the teacher's criteria, represented by $max_{l\in\{0,1\}}\sum_{t=1}^H r(s_t^l, a_t^l) < \delta_{skip}$, the teacher chooses not to provide a preference annotation.

As true rewards vary across environments, the B-Pref benchmark uses an adaptive threshold $\delta_{adapt}$ for either $\delta_{equal}$ or $\delta_{skip}$. This threshold is calculated as:

$$
\delta_{\text{adapt}}(\epsilon_{\text{adapt},t}) = \frac{H}{T} R_{\text{avg}}(\pi_t)\epsilon_{\text{adapt}},
\tag{4}
$$

where $t$ represents the current time step, $\epsilon_{adapt} \in [0,1]$ is a hyperparameter governing $\delta_{adapt}$, $T$ denotes the episode length, and $R_{avg}(\pi_t)$ is the average return of current policy $\pi_t$.

## 3 Proposed method

In this section, we describe our method MCP in detail. Figure 1 illustrates the overall architecture of MCP. Our main idea is to enhance the agent's robustness against the corrupted preferences and improve feedback efficiency by applying mixup augmentation to labeled preference data. Additionally, MCP can be seamlessly integrated with existing preference-based RL methods, highlighting its potential as a complementary enhancement to established techniques.

Mixup generates new instances through the interpolation between two instances, where each instance consists of single input and single output (Zhang et al., 2018). However, directly applying mixup to preference-based RL is intractable because of the nature of a single instance (preference) which consists of a pair of behavior trajectories as input alongside a corresponding preference label. In light of this, we propose MCP, a modified method for mixing two preferences. Initially, we define the mixup between two trajectory segments, $\sigma^i$ and $\sigma^j$, as generating an element-wise convex combination as follows:

$$
\tilde{\sigma} = \text{SegMix}(\sigma^i, \sigma^j, \lambda) = \left\{(\lambda s_t^i + (1-\lambda)s_t^j, \lambda a_t^i + (1-\lambda)a_t^j)\right\}_{t=1}^H,
\tag{5}
$$

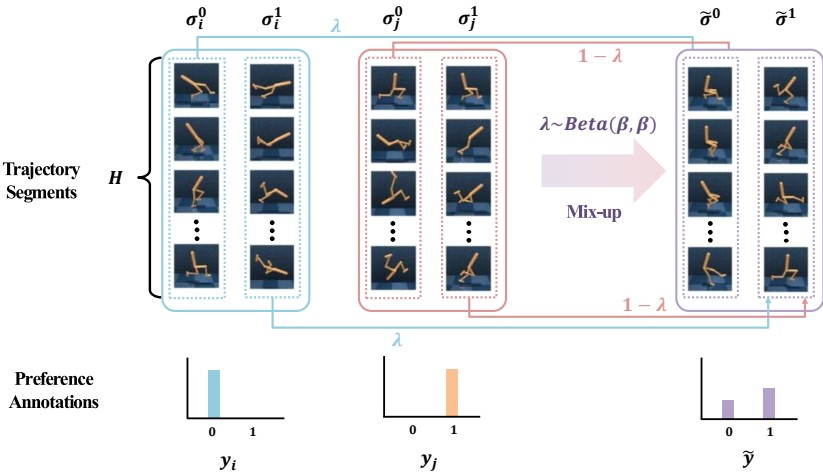

Figure 1: An overall architecture of MCP. We blend the segments at the corresponding indices within two preference data using mixup ratio $\lambda$. The mixed label distribution is generated by interpolating two preference labels with same ratio.

where $\lambda$ follows the Beta distribution $Beta(\beta, \beta)$. Herein, $\beta \in (0, \infty)$ is user-defined hyperparameter. We sample a pair of $p_i = (\sigma_i^0, \sigma_i^1, y_i)$ and $p_j = (\sigma_j^0, \sigma_j^1, y_j)$, where $p_k$ signifies the $k$th entry $(\sigma_k^0, \sigma_k^1, y_k)$ in the labeled preference data $D$. In contrast to conventional methods, we acknowledge the presence of corrupted preferences in $D$. We then generate new preference $\tilde{p} = (\tilde{\sigma}^0, \tilde{\sigma}^1, \tilde{y})$ through component-wise convex combination of $p_i$ and $p_j$ as follows:

$$\tilde{\sigma}^l = \text{SegMix}(\sigma_i^l, \sigma_j^l, \lambda) \quad \forall l \in \{0, 1\}, \tag{6}$$

$$\tilde{y} = \lambda y_i + (1 - \lambda) y_j. \tag{7}$$

The new preference $\tilde{p}$ generated by mixup is then used to train the reward estimator in the same manner as existing labeled preferences. It should be noted that we do not use additional tasks or networks to detect or mitigate corrupted preferences. MCP offers two distinct advantages: (1) In line with the VRM principle, MCP regularizes the memorization, thereby enhancing robustness against corrupted preferences. (2) MCP improves feedback efficiency by enabling the limitless augmentation of new preference data, even from a small amount of feedback instances.

In each iteration, MCP begins by sampling $\lambda$ from the Beta distribution. Using the sampled $\lambda$, we then generate a new batch, denoted as $\tilde{P}$, through mixing the original preference batch $P$ and a batch of indices that have been randomly shuffled $P'$, following Equations 6 and 7. The reward estimator $r_\psi$ is subsequently trained by minimizing an objective that takes into account both $P$ and $P'$. This objective is given by:

$$L_\psi^{MCP} = E_{(\sigma^0, \sigma^1, y) \sim P, (\tilde{\sigma}^0, \tilde{\sigma}^1, \tilde{y}) \sim \tilde{P}}[L^{Reward}(\sigma^0, \sigma^1, y) + L^{Reward}(\tilde{\sigma}^0, \tilde{\sigma}^1, \tilde{y})], \tag{8}$$

where $L^{Reward}$ is consistent with the definition in Equation 2. The pseudo-code for MCP is described in Algorithm 1 in Appendix B.

## 4 EXPERIMENTS

We design our experiments to address two key questions: (1) Can MCP enhance the performance of previous preference-RL methods when confronted with scenarios involving corrupted preference? (2) Can MCP perform well even in situations with limited data in terms of feedback efficiency?

**Environment and evaluation.** We evaluated MCP across three locomotion tasks from DeepMind Control Suite (DMControl) (Tassa et al., 2018; Tunyasuvunakool et al., 2020), and six robotic manipulation tasks from Meta-world (Yu et al., 2020) (See Figure 8 in Appendix C). Similar to previous studies, we used a scripted human teacher that provides feedback to the agent, described in 2.2. To

consider human irrationalities, we concentrated on the four annotation strategies by adjusting the hyperparameters in Equation 3 and Equation 4 as follows:

- Oracle: $(\eta_{stoc} \to \infty, \eta_{myopic} = 1, \epsilon = 0, \delta_{equal} = 0, \delta_{skip} = 0)$
- Mistake: $(\eta_{stoc} \to \infty, \eta_{myopic} = 1, \epsilon = 0.1, \delta_{equal} = 0, \delta_{skip} = 0)$
- Equal: $(\eta_{stoc} \to \infty, \eta_{myopic} = 1, \epsilon = 0, \delta_{equal} = \delta_{adapt}(\epsilon_{adapt} = 0.1, t), \delta_{skip} = 0)$
- Stochastic: $(\eta_{stoc} = 1, \eta_{myopic} = 1, \epsilon = 0, \delta_{equal} = 0, \delta_{skip} = 0)$

For our experiments, we adopted the following evaluation metrics: true average episode rewards over ten episodes for DMControl tasks and the success rate over ten episodes for Meta-world tasks. All experiments were conducted over 1 million environment steps. The results are shown in the learning curves. Here, the solid line signifies the mean, while shaded regions correspond to the standard deviation across ten independent runs. In addition, we measured the final performance of each algorithm using the average over the last ten evaluation scores, shown by dotted lines.

**Implementation.** Incorporating MCP into existing preference-based RL algorithms is straightforward. It simply requires replacing the reward learning procedure with Algorithm 1 in Appendix B. For our study, we selected the well-known method PEBBLE (Lee et al., 2021b) as our base algorithm. Because PEBBLE utilizes the SAC (Haarnoja et al., 2018b) algorithm for its policy and critic network learning, we also used SAC—trained using ground truth rewards—as a standard to assess the effectiveness of both PEBBLE and our MCP-enhanced approach. Consistently across experiments, we retained the same hyperparameters as those used in the original SAC and PEBBLE studies. To acquire informative queries, we used an ensemble of three reward estimators and applied a disagreement-based sampling (Christiano et al., 2017) (see Appendix C for hyperparameters and more details). For MCP, we set $\beta = 8$ for DMControl environments and $\beta = 0.5$ for Meta-world environments.

## 4.1 BENCHMARK TASK RESULTS WITH MISTAKE CORRUPTION

Mistake scenario is the most challenging corruption strategy, consistently showing significant performance degradation when compared to the Oracle scenario (Lee et al., 2021a). To demonstrate robustness of MCP, we compared PEBBLE with its integration of MCP in the Mistake scenario using the same quantities of feedback. In addition, we conducted experiments under the Oracle scenario for both algorithms, establishing an upper performance benchmark for each.

**DMControl results.** We selected three intricate environments from DMControl, each designed to represent distinct locomotion tasks: Walker-walk, Cheetah-run, and Quadruped-walk. These environments are designed to challenge the agent in advancing forward by carefully manipulating posture and velocity based on physical principles. Figure 2 shows the results of both PEBBLE and its integrated counterpart with MCP under Oracle and Mistake scenarios. It is evident from the figure that MCP has a remarkable influence, leading to substantial performance enhancements across all three environments when dealing with Mistake-induced corruption. Of particular interest is the observation that MCP, trained with Mistake corruptions, either equals or even outperforms PEBBLE trained within Oracle scenarios in Walker and Cheetah environments. Furthermore, we found a noteworthy discovery: with only 100 clean preferences (Oracle), MCP achieves performance levels that are comparable to SAC trained with ground truth rewards on Walker task, whereas PEBBLE necessitates about four times more feedback, as reported in Lee et al. (2021b). This encouraged us to delve deeper into evaluating feedback efficiency in Section 4.2

**Meta-world results.** Meta-world offers an extensive selection of environments designed for intricate robotic manipulation tasks, and we specifically targeted six tasks: Window Open, Door Unlock, Button Press, Door Open, Drawer Open, and Sweep Into. The learning curves depicted in Figure 3 showcase the performance of both algorithms under equivalent feedback quantities. As in the DM-Control experiments, our observations were consistent: MCP effectively mitigates the performance degradation resulting from mistakenly flipped preferences, exhibiting a substantial gap of success rate when compared to PEBBLE in the Mistake scenario. Significantly, across the four environments —Door Unlock, Button Press, Drawer Open, and Sweep Into— PEBBLE exhibits a significant decline in success rate within the Mistake scenario compared to the Oracle scenario. However, MCP,

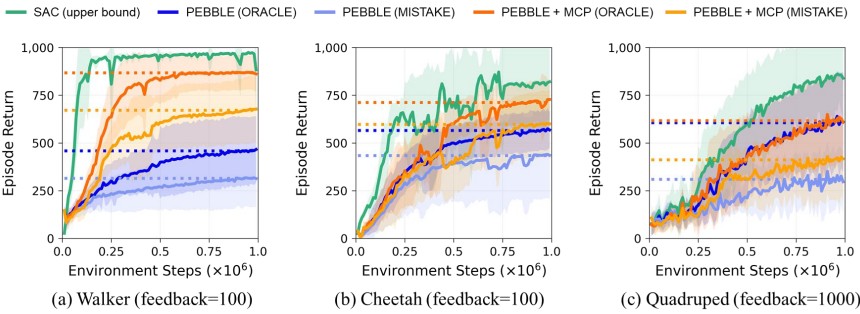

Figure 2: Learning curves on three locomotion tasks as measured on the ground truth reward under Oracle and Mistake scenarios.

when trained under the Mistake scenario, maintains performance levels on par with the Oracle scenario, thereby underscoring its robustness. Moreover, we observed that MCP trained with clean feedback (Oracle) achieves nearly 100% success rate within half the duration of training time in the Window Open and Door Open tasks, showing its high feedback efficiency. In comparison, PEBBLE requires approximately four and six times more feedback instances than MCP to achieve equivalent score at the end of training, as reported in Lee et al. (2021b).

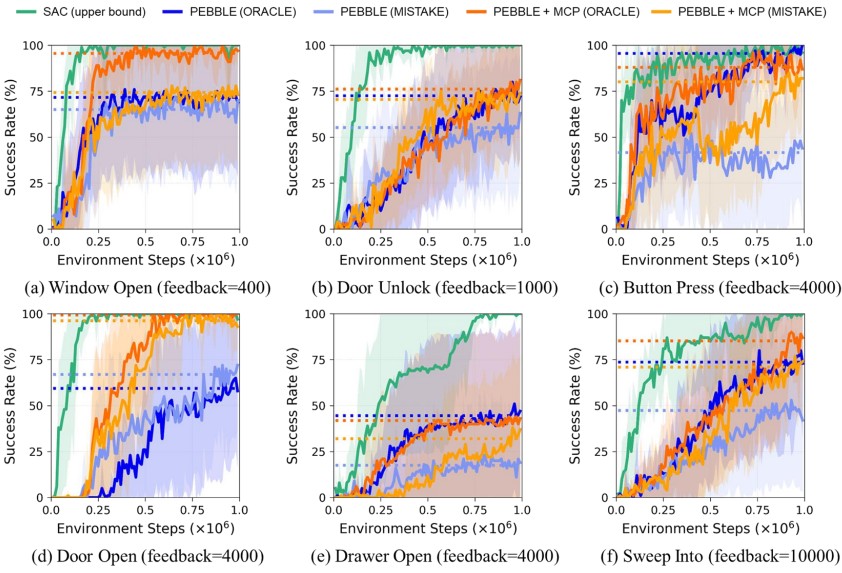

Figure 3: Learning curves on six robotic manipulations tasks as measured on the success rate under Oracle and Mistake scenarios.

## 4.2 FURTHER ANALYSIS

**Quality of estimated rewards.** We assessed the capability of the reward estimator, trained using MCP, to capture user intent even when faced with corrupted preferences. For our analysis, we employed Spearman's rank correlation, which is well-suited for consistent non-linear associations. We randomly chose 100 episodes and then calculated the correlation between ground truth and estimated rewards for two different reward estimators: PEBBLE (Mistake), and PEBBLE + MCP (Mistake). A higher correlation suggests that the reward estimator is more adept at distinguishing among the quality of trajectories, even when some preferences might have been mistakenly flipped. Figure 4 displays the mean and standard deviation of the correlation coefficients over ten runs. As evident from Figure 4, MCP consistently outperforms PEBBLE in terms of correlation values under

the Mistake scenario. This highlights the proficiency of MCP in mitigating potential misconceptions about user intent due to corrupted preferences, aligning with our findings in Section 4.1.

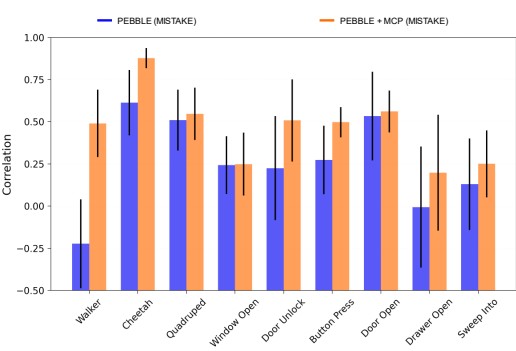

Figure 4: Grouped bar plot illustrating the Spearman rank correlation coefficient calculated over 100 randomly selected episodes.

**Investigation of feedback efficiency.** To further explore feedback efficiency of MCP, we conducted comprehensive experiments encompassing varying feedback quantities, comparing PEBBLE with its integration with MCP under the Oracle scenario (Figure 5). Specifically, we selected one DMControl task, Walker, and conducted experiments using feedback quantities of 50, 100, 500, and 1,000. Additionally, we extended this investigation to include two Meta-world tasks: Door Open and Sweep Into. For these tasks, we used feedback quantities of 1,000, 2,000, 4,000, and 5,000 for Door Open, and feedback quantities of 2,000, 4,000, 5,000, and 10,000 for Sweep Into. Performance enhanced by MCP is noteworthy, particularly when the number of feedback instances is small. Notably, in the Walker and Door Open tasks, the performance disparity between using and not using MCP gradually diminishes with an increase in feedback quantity. A similar trend is observed in the Sweep Into task, where a significant performance gap, particularly around 4,000 feedback instances, gradually decreases. This reduction can be attributed to the increased availability of comprehensive information from feedback, which gradually mitigates the initial performance gap originating from insufficient feedback quantities (Liu et al., 2022). These findings underscore that MCP enhances feedback efficiency by generating feasible trajectory segment pairs and corresponding preference labels.

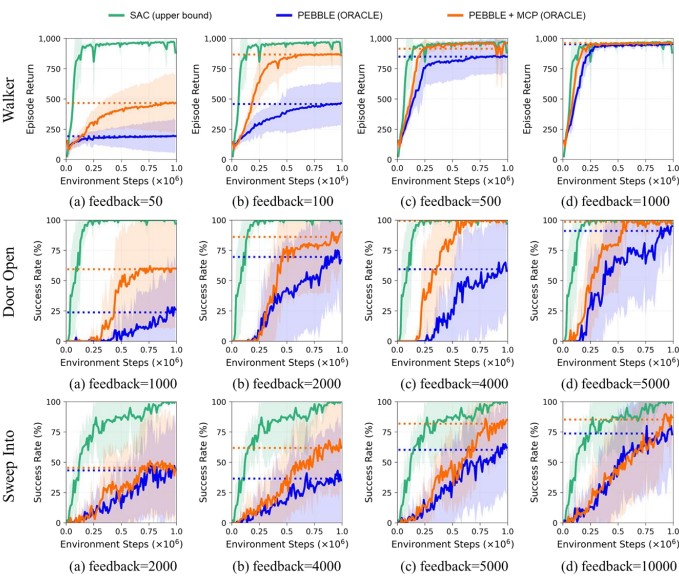

Figure 5: Learning curves on Walker (first row), Door Open (second row) and Sweep Into (third row) with varying feedback quantities under the Oracle scenario.

**Complementary effect analysis.** We examined the synergistic potential of integrating MCP with other preference-based RL approaches, focusing on SURF, a semi-supervised reward learning approach (Park et al., 2022). Similar to MCP, SURF improved feedback efficiency when coupled with PEBBLE. We investigated whether MCP can further improve both the robustness and feedback ef-

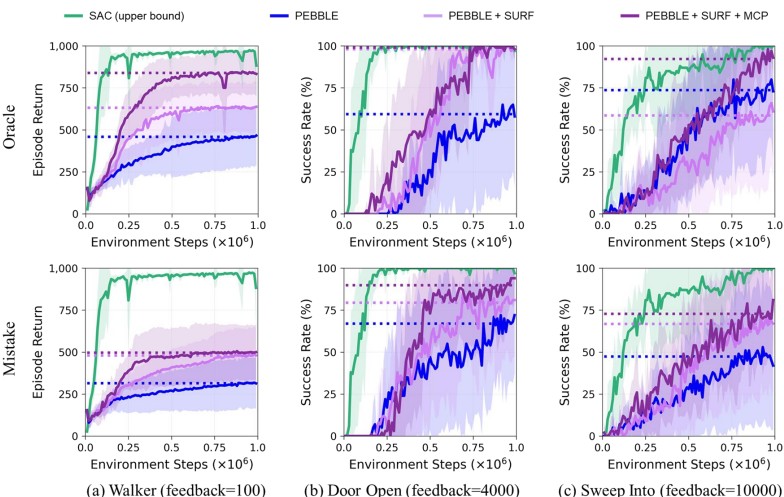

Figure 6: Learning curves on Walker, Door Open, and Sweep Into when integrated with SURF. The first row represents the Oracle scenario and the second row represents the Mistake scenario.

ficiency with SURF across three environments. The hyperparameters used for SURF are described in Appendix C. Figure 6 demonstrates that MCP significantly enhances the SURF's performance, achieving performance comparable to SAC trained with ground truth reward in the Oracle scenario. In addition, the combination of MCP and SURF in the Mistake scenario performs similarly to SURF in the Oracle scenario, highlighting ability of MCP to mitigate the effects of corrupted preferences.

**Other corruption strategies.** We tested MCP against various corruption strategies, namely Equal and Stochastic. Across different forms of human irrationality, our method demonstrated consistent effectiveness. Detailed results and analyses are described in Appendix D.1. We did not explore the Skip and Myopic strategies because their performance degradations are less pronounced, as previously observed in the B-Pref Benchmark (Lee et al., 2021a).

**Varying mistake strength.** To further investigate the robustness of MCP, we extended our experiments within the Mistake scenario to include the Walker, Door Open, and Sweep Into tasks. We adjusted the mistake strength factor $\epsilon$ incrementally from 0 to 0.2 in steps of 0.05, while other hyperparameters were held constant as in the original Mistake scenario. Across all $\epsilon$ values, MCP consistently surpassed PEBBLE. These findings are shown in Appendix D.2.

**Hyperparameter investigation.** We conducted a series of experiments to determine the optimal mixup hyperparameter $\beta$ for each domain under the Mistake scenario. We experimented with different $\beta$ values (0.5,1,2,4,8). More in-depth results are described in Table 6 of Appendix D.3.

**Comparison with other methods** To ensure a comprehensive comparison, we evaluated several preference-based RL methods. The results include SURF in both Oracle and Mistake scenarios, adding six more environments not shown in Figure 6. We also re-implemented RUNE (Liang et al., 2022) and MRN (Liu et al., 2022). Further details of implementation and results can be found in Appendix D.4.

**Ablation with original preference data.** We conducted an ablation study to determine if utilizing original preference data for reward estimator learning significantly impacts MCP's performance. The learning curves and findings are available in Appendix D.5

**Effect of calibrating overconfidence.** Overconfidence in reward estimator learning can lead to a significant divergence of reward estimations between two trajectories compared to the ground-truth. In Appendix D.6, we demonstrate that the reward estimator trained with MCP is better calibrated than the one used in PEBBLE.

## 5 RELATED WORK

**Preference-based reinforcement learning.** Many studies have been conducted to train RL agents with human feedback. Christiano et al. (2017) proposed a general framework of preference-based RL and introduced ensemble-based sampling for selecting highly informative queries. To collect diverse trajectories, several methods, such as unsupervised pre-training (Lee et al., 2021b) and uncertainty-based exploration (Liang et al., 2022) have been proposed. Another area of research has focused on addressing the scarcity of labeled preferences through the integration of imitation learning (Ibarz et al., 2018) or the adoption of self- or semi-supervised learning (Park et al., 2022; Metcalf et al., 2022). Other approaches have aimed to improve the representational capacity of the reward estimator by considering information of previous transitions with a non-Markovian reward estimator (Kim et al., 2023) or incorporating bi-level optimization using the performance of Q-function (Liu et al., 2022). Although previous methods have made significant progress in preference-based RL, they have mainly considered scenarios with perfect preference annotation. However, mistakenly mislabeled preferences can significantly deteriorate the performance of preference-based RL (Lee et al., 2021a). Concurrent with our work, Xue et al. (2023) stabilizes reward learning by employing KL divergence regularization and confidence-based ensemble to address the corrupted preferences with crowd-sourced annotations. In this work, we address the performance degradation caused by the corrupted preference problem by applying mixup augmentation to labeled preferences.

**Learning with corrupted labels.** Several methods have been proposed to tackle the challenge of the label corruption. Jiang et al. (2018) introduced MentorNet,an extra network identifying clean instances based on the loss histories of entire dataset. However, the necessity of clean dataset limits the applicability of MentorNet (Han et al., 2018; Arazo et al., 2019). To address this issue, several methods used bi-modal mixture model to determine the sample is clean or not (Arazo et al., 2019; Li et al., 2020). Another approaches built co-assisting twin networks, wherein each network selects small-loss samples (considered as corrupted) to train its counterpart (Han et al., 2018; Yu et al., 2019b; Li et al., 2020). Meanwhile, To reduce memorization and improve the generalization of model, mixup (Zhang et al., 2018) augmentation was introduced, demonstrating outstanding robustness against corrupted labels and adversarial noises. Mixup offers several notable advantages. First, it eliminates the need for separate clean dataset (Jiang et al., 2018; 2020). Second, it avoids intricate mechanisms for selecting clean data such as periodic mixture model fitting, reciprocal training of twin networks, and storing loss values of the entire training dataset. In addition, mixup is highly data-efficient, so that many studies have proposed variants of mixup to overcome data deficiency in various fields (Lin et al., 2021; Zhang et al., 2020; Guo et al., 2020). In this paper, we applied mixup to preference-based RL to draw its robustness and data efficiency.

## 6 CONCLUSION AND FUTURE WORK

In this study, we introduce MCP, a preference-based RL approach that utilizes mixup augmentation to enhance model robustness and feedback efficiency. While conventional mixup techniques are effective for individual instances, our method addresses the challenge of mixing pairs of preferences—each consisting of two trajectory segments and an associated preference label. To this end, we propose a modified mixup strategy tailored for preference-based RL, allowing for convex combinations of trajectory segments and preference data pairs. Our experiments demonstrate the outstanding robustness of MCP across various tasks, effectively mitigating the performance degradation arising from corrupted preferences. Notably, MCP also significantly improves feedback efficiency, outperforming previous methods in terms of both speed and the number of feedback instances required. We further highlight the compatibility of MCP with various preference-based RL methods, thereby enhancing their robustness and feedback efficiency. We believe that MCP has great potential for enhancing preference-based RL in complex real-world scenarios, even when limited and inaccurate preferences exist. Several promising avenues for future research of MCP are worth exploring. First, it is important to investigate the effectiveness of MCP in discrete action spaces. Although MCP's reward learning is not inherently limited by the type of action space, experimental validation of its effectiveness remains important. This expansion will make MCP more versatile and applicable across a broader spectrum of environments. Second, introducing an adaptive approach to control the mixup hyperparameter is also an interesting direction, reducing the effort required to manually determine an appropriate value.

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

## A    ADDITIONAL PRELIMINARIES

**Mixup augmentation.**    The mixup technique generates new data points by linearly interpolating two input data points and their corresponding labels. This process creates a vicinal distribution, which enhances generalization and improves robustness against corrupted data (Zhang et al., 2018). Additionally, mixup is known to enhance confidence calibration by regularizing overconfidence and providing a more reliable indication of prediction uncertainty (Thulasidasan et al., 2019). Given two data points, $(x_i, y_i)$ and $(x_j, y_j)$, the augmented sample $(\tilde{x}, \tilde{y})$ is formulated as:

$$\tilde{x} = \lambda x_i + (1 - \lambda) x_j, \tag{9}$$

$$\tilde{y} = \lambda y_i + (1 - \lambda) y_j, \tag{10}$$

where $\lambda \sim Beta(\alpha, \alpha)$ and $\alpha$ is a non-negative hyperparameter. The hyperparameter $\alpha$ controls the level of mix, with $\lambda$ tending towards 0.5 as $\alpha$ increases. Conversely, a lower $\alpha$ leads to a mix that largely favors one of the original data points.

## B    ALGORITHM DETAILS

In this section, We detail the procedure of MCP. We adopted PEBBLE as our backbone algorithm and modified its reward learning procedure, as outlined in Algorithm 1. Figure 7 illustrates the mixed trajectory, which is the output of SegMix in Equation 6.

---

**Algorithm 1** MCP: Mixing Corrupted Preferences

---

1: **Require:** Hyperparameters: parameter of beta distribution $\beta$
2: **Input:** Set of labeled preference data $D$
3: **for** each gradient step **do**
4:      Sample labeled batch $P = \{(\sigma_i^0, \sigma_i^1, y_i)\}_{i=1}^{N_B} \sim D$
5:      Sample $\lambda \sim Beta(\beta, \beta)$                                          ▷ sample mixup ratio
6:      $P' = \text{RandomShuffle}(P) = \{(\sigma_j^0, \sigma_j^1, y_j)\}_{j=1}^{N_B}$                  ▷ randomly shuffle indices of $P$
7:      $\tilde{P} = \{\}$                                                        ▷ empty batch
8:      **for** $N_B$ iterations **do**                                  ▷ $i$ is index of $P$, $j$ is index of $P'$
9:          $\tilde{\sigma}^0 = \text{SegMix}(\sigma_i^0, \sigma_j^0, \lambda)$      ▷ Equation 6: mixing the first segments of each instance
10:          $\tilde{\sigma}^1 = \text{SegMix}(\sigma_i^1, \sigma_j^1, \lambda)$      ▷ Equation 6: mixing the second segments of each instance
11:          $\tilde{y} = \lambda y_i + (1 - \lambda) y_j$                ▷ Equation 7: mixing preference labels of each instance
12:          Append $(\tilde{\sigma}^0, \tilde{\sigma}^1, \tilde{y})$ to $\tilde{P}$
13:      **end for**
14:      Optimize $L_\psi^{MCP}$ in Equation 8 with respect to $\psi$
15: **end for**

---

## C    EXPERIMENTAL DETAILS

**Training details.**    The SAC agent is trained with ground-truth reward function, and Table 1 provides a full list of the hyperparameters for SAC. For the reward model used in preference-based algorithms, we used an ensemble estimator comprising three MLP models, each consisting of three layers with 256 hidden units and leaky ReLU activation. The output of each ensemble member is bounded within [-1, 1] range using the tanh function. These models are trained to minimize the binary cross-entropy loss using Adam optimizer (Kingma & Ba, 2015), with the batch size of 128 and the learning rate of 0.0003. The hyperparameters for PEBBLE align with the settings in B-Pref benchmark (Lee et al., 2021a), and are detailed in Table 2. Most of hyperparameters for SURF are consistent with those of PEBBLE, and the remained hyperparameters are outlined in Table 3. The only hyperparameter required for MCP is $\beta$, which is described in implementation paragraph in Section 4. The determination of $\beta$ was executed through a hyperparameter search conducted in Appendix D.3.

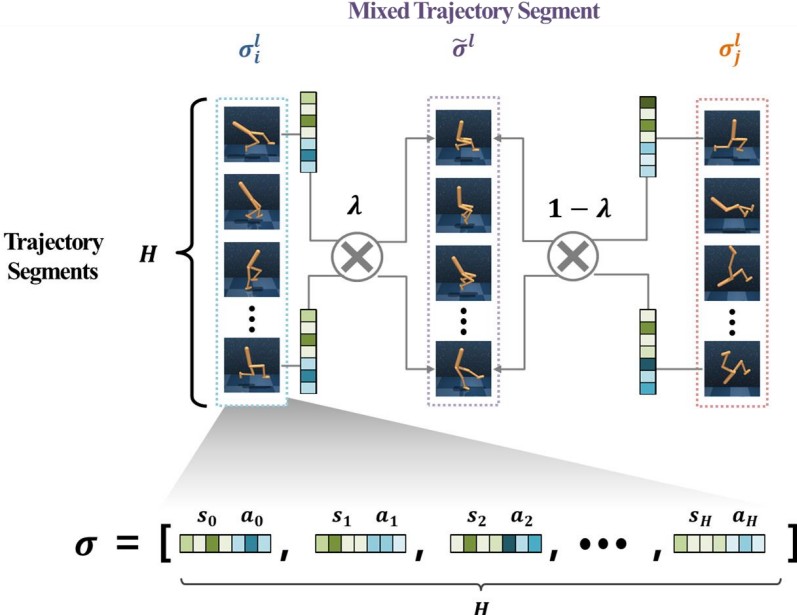

Figure 7: Generating a mixed trajectory using SegMix. Each component in the state-action sequence of the mixed segment is formed by interpolating corresponding components from two different trajectories at the same time index.



Figure 8: Rendered images of environments we tested on. The first row shows three locomotion tasks of DMControl and the other rows show six robotic manipulation tasks of Meta-world.

**Sampling scheme.** We employed a disagreement-based sampling to acquire informative queries using ensemble estimator in all experiments (Christiano et al., 2017; Lee et al., 2021b; Park et al., 2022; Liang et al., 2022). First, we uniformly sample an initial batch of $N_{init}$ pairs of trajectory segments from replay Buffer $B$. Subsequently, we measured the uncertainty as variance across the ensemble predictions $\left\{P_{\psi_i}(\sigma^1 \succ \sigma^0)\right\}_{i=1}^{N_{en}}$, and then selected $N_{query}$ pairs which have high uncertainty. Consistent with previous methods, we set $N_{init}$ as ten times the value of $N_{query}$.

Table 1: Hyperparameters of SAC algorithm.

| Hyperparameter | Value | Hyperparameter | Value |
|---|---|---|---|
| Init temperature $\alpha$ | 0.1 | Hidden units per each layer | 1024 (DMControl) |
| Learning rate for $\theta, \phi$ | 0.0005 (Walker, | | 256 (Meta-world) |
| | Cheetah) | # of layers | 2 (DMControl) |
| | 0.0001 (Quadruped) | | 3 (Meta-world) |
| | 0.0003 (Meta-world) | Batch size for $\theta, \phi$ | 1024 (DMControl) |
| Target critic update frequency | 2 | | 512 (Meta-world) |
| Optimizer | Adam (Kingma & Ba, 2015) | Critic EMA $\tau$ | 0.005 |
| Optimizer temperatures $(\beta_1, \beta_2)$ | (0.9,0.999) | Discount factor $\gamma$ | 0.99 |

Table 2: Hyperparameters of PEBBLE algorithm.

| Hyperparameter | Value | Hyperparameter | Value |
|---|---|---|---|
| Batch size for $\psi$ | 128 | Length of segment | 50 (DMControl) |
| Total feedbacks/ | 100/10 (Walker, Cheetah) | | 25 (Meta-world) |
| # of queries per session | 1000/100 (Quadruped) | Frequency of feedback | 20000 (Walker, |
| | 400/10 (Window Open) | | Cheetah) |
| | 1000/10 (Door Unlock) | | 30000 (Quadruped) |
| | 4000/20 (Button Press, Door Open, | | 5000 (Meta-world) |
| | Drawer Open) | Data collection steps | 1000 |
| | 10000/50 (Sweep Into) | Pre-training steps | 9000 |

Table 3: Hyperparameters of SURF algorithm.

| Hyperparameter | Value |
|---|---|
| Unlabeled batch ratio $\mu$ | 4 |
| Loss weight for unlabeled data | 1 |
| Segment length before cropping | 60 (DMControl), |
| | 35 (Meta-world) |
| Mix/Max length of cropped segment | [45, 55] (DMControl), |
| | [20, 30] (Meta-world) |
| Threshold $\tau$ | 0.999 (Cheetah, Window Open, Sweep Into), |
| | 0.99 (Others) |

Table 4: Hyperparameters of RUNE algorithm.

| Hyperparameter | Value |
|---|---|
| Initial weight for intrinsic reward | 0.05 |
| Decaying rate $\rho$ | 0.00001 |

Table 5: Hyperparameters of MRN algorithm.

| Hyperparameter | Value |
|---|---|
| Meta-step frequency | 1000 (Walker), |
| | 3000 (Quadruped), |
| | 10000 (Door Open, Sweep Into), |
| | 5000 (Others) |

# D ADDITIONAL EXPERIMENTAL RESULTS

## D.1 BENCHMARK TASK RESULTS UNDER OTHER CORRUPTION STRATEGIES

**Equal corruption results.** In the context of equal corruption, as illustrated in Figure 9, both algorithms exhibit significant improvements when compared to the Oracle scenario, especially within specific environments. Notably, in the Quadruped task, MCP marginally outperforms SAC trained with ground truth reward, even with just 1,000 feedback instances, while also showing a lower standard deviation. These findings reveal that the defined reward function may not always perfectly align with the intended task success. Furthermore, MCP has the potential to efficiently address user-intended tasks with a small amount of feedback instances, even in the presence of minor in-

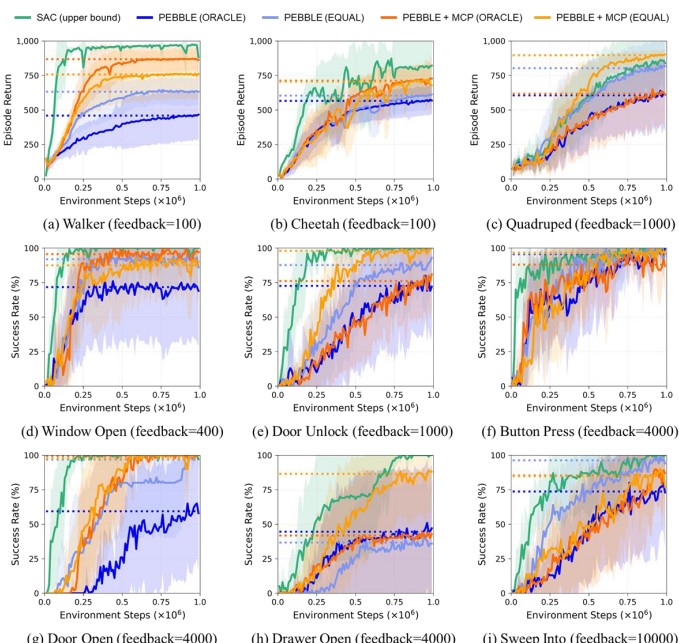

Figure 9: Learning curves on three locomotion tasks (first row) and six robotic manipulation tasks (second and third rows) under Oracle and Equal scenarios.

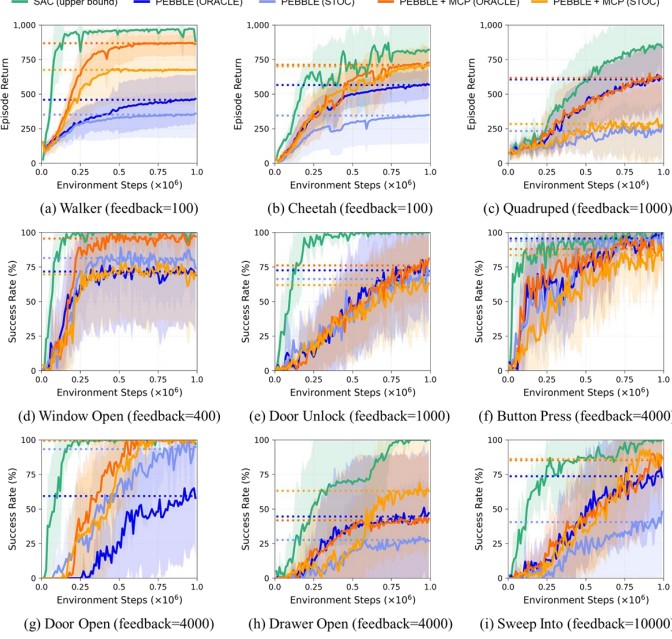

Figure 10: Learning curves on three locomotion tasks (first row) and six robotic manipulation tasks (second and third rows) under Oracle and Stochastic scenarios.

accuracies in discerning superiority. In all DMControl tasks, consistent with the observation in the Mistake scenario, MCP improved the robustness over PEBBLE under the Equal scenario (first row). Within Meta-world tasks (second and third rows), the enhancement by MCP stands out in three environments: Door Unlock, Door Open, and Drawer Open. Notably, MCP's performance improvement in Drawer Open is remarkable. It achieves an almost 90% success rate in the Equal

scenario, doubling its Oracle scenario performance. In contrast, PEBBLE exhibits a slight drop in performance.

**Stochastic corruption results.** Figure 10 presents the learning curves of both algorithms with Oracle and Stochastic scenarios. In DMControl tasks (first row), consistent with the aforementioned scenarios, MCP mitigates the negative impact of corrupted preferences. Furthermore, MCP trained with corrupted preferences still surpasses PEBBLE trained with clean annotations in two of the three locomotion environments. Within Meta-world environments (second and third rows), both MCP and PEBBLE exhibit comparable high success rates within the Stochastic scenario. However, PEBBLE significantly suffers from the stochastic preference decision in Drawer Open and Sweep Into. Conversely, MCP maintains consistent performance even in the presence of stochasticity in the Sweep Into task. Notably, within the Drawer Open environment, MCP's Stochastic scenario performance considerably improves compared to the Oracle scenario. These observations support the earlier conclusion that MCP can handle tasks with limited feedback, which might be inaccurate. Interestingly, certain environments indicate that the deterministic teacher (Oracle) does not invariably surpass the Equal and Stochastic corruption scenarios. In these scenarios, the corrupted preferences do not always hinder preference-based RL methods. Nevertheless, the more demanding Mistake scenario (Figures 2 and Figure 3) consistently exhibits adverse effects on reward estimator learning, highlighting the indispensability of MCP to enhance robustness.

### D.2 DETERIORATION ON VARYING MISTAKE STRENGTH

We extended our experiments under the Mistake scenario with one DMControl environment (Walker) and two Meta-world environments (Door Open and Sweep Into), by varying the mistake strength factor $\epsilon$. Figure 11 shows that the performance of MCP within the Walker task remains consistently comparable to that of PEBBLE under the Oracle scenario, even when 20% of preference labels are mistakenly flipped. In the Door Open task, MCP still achieves a 100% success rate up until the mistake strength reaches 10%. Even at a mistake strength of 20%, MCP continues to maintain a success rate over 75%, whereas PEBBLE experiences a significant deterioration, resulting in a success rate of approximately 25%. In the context of the Sweep Into task, MCP demonstrates its robustness up to a mistake strength of 10%, outperforming PEBBLE which begins to display signs of decline. However, the inherent complexity of the Sweep Into task causes both algorithms to exhibit reduced performance at higher levels of mistake strengths

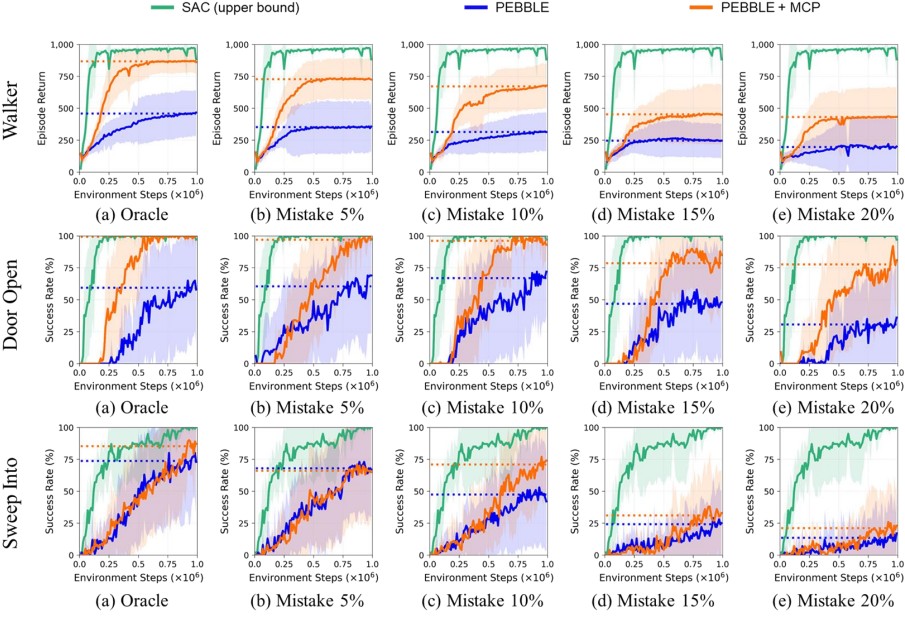

Figure 11: Learning curves on Walker (first row), Door Open (second row) and Sweep Into (third row) with varying mistake strength.

### D.3  Hyperparameter investigation

We conducted a hyperparameter search to determine the optimal mixup hyperparameter $\beta$. Table 6 represents the final performance of MCP under the Mistake scenario, using varying mixup hyper-parameters. Recognizing the sensitivity of average-based assessments to outliers and scale-related effects, we adopted a rank-based assessment approach to determine the value of $\beta$ for each domain. The average rank values are indicative of the relative performance rankings across different $\beta$ values for each domain. Based on Table 6, we determined $\beta = 8$ in DMControl environments and $\beta = 0.5$ in Meta-world environments across all experiments.

Table 6: Comparison of final performance of MCP under Mistake scenario with varying mixup hyperparameters.

| Domain | Task | $\beta = 0.1$ | $\beta = 0.5$ | $\beta = 1$ | $\beta = 2$ | $\beta = 4$ | $\beta = 8$ |
|---|---|---|---|---|---|---|---|
| DMCotrol Episode Return | Walker (feedback=100) | 397.1 | 493.3 | 582.1 | 421.7 | 510.6 | 671.6 |
| | Cheetah (feedback=100) | 658.7 | 621.0 | 551.5 | 573.9 | 607.1 | 597.4 |
| | Quadruped (feedback=1000) | 316.1 | 353.3 | 257.6 | 359.9 | 511.9 | 421.7 |
| Average Rank | | 4.0 | 3.3 | 4.7 | 4.3 | **2.3** | **2.3** |
| Meta-world Success Rate (%) | Window Open (feedback=400) | 67.1 | 74.3 | 47.5 | 60.3 | 61.5 | 83.4 |
| | Door Unlock (feedback=1000) | 47.2 | 70.5 | 62.7 | 60.7 | 59.6 | 66.4 |
| | Button Press (feedback=4000) | 78.9 | 80.2 | 72.7 | 61.6 | 88.9 | 71.1 |
| | Door Open (feedback=4000) | 87.6 | 96.2 | 97.8 | 98 | 82.4 | 91.1 |
| | Drawer Open (feedback=4000) | 68.6 | 32.0 | 47.3 | 47.9 | 59.9 | 56.0 |
| | Sweep Into (feedback=10000) | 66.4 | 71.0 | 64.3 | 49.0 | 74.6 | 66.3 |
| Average Rank | | 3.5 | **2.7** | 4.2 | 4.3 | 3.2 | 3.2 |

### D.4  Comparison with other Preference-based RL Methods

To conduct a thorough comparison, we evaluated MCP against other preference-based RL methods under both Oracle and Mistake scenarios. We extended our analysis with (Park et al., 2022) to include six additional environments not covered in Figure 6. We also included results for RUNE (Liang et al., 2022) and MRN (Liu et al., 2022). All experiments were conducted over identical ten random seeds. For each algorithm, we used their official implementations and configured hyperparameters as specified in the original papers (see Appendix C for details).

Figures 12 and 13 show the learning curves for the Oracle and Mistake scenarios, respectively. MCP consistently outperforms other methods in most environments under the Mistake scenario. Additionally, we used *rliable* library (Agarwal et al., 2021) to report SAC-normalized interquartile mean (IQM) and optimality gap (OG) for each domain (refer to Figure 14). IQM represents the middle 50% performance across all runs and higher values are preferable. OG measures the divergence of the agent's performance from the optimal level (SAC in our study), with lower values being preferable. Both metrics are robust against performance variability and outliers. The learning curves and these diverse metrics together show that MCP significantly surpasses other baselines.

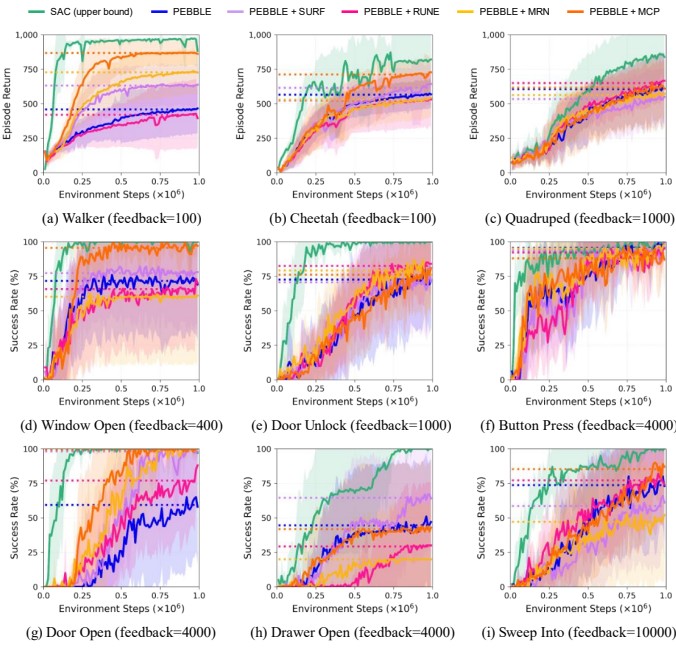

Figure 12: Learning curves on three locomotion tasks (first row) and six robotic manipulation tasks (second and third rows) under Oracle scenarios.

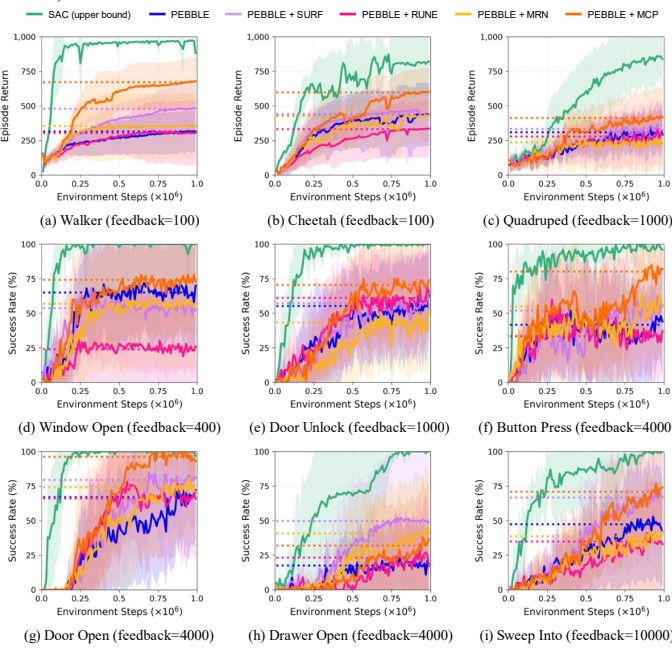

Figure 13: Learning curves on three locomotion tasks (first row) and six robotic manipulation tasks (second and third rows) under Mistake scenarios.

## D.5 ABLATION WITH USING ORIGINAL PREFERENCES

We conducted an ablation study to assess how using original preference data affects the performance of MCP. Specifically, we focused on three environments (Walker, Door Open, and Sweep Into) and used only the second term of Equation 2 across ten runs. Figure 15 shows that MCP, when trained solely with mixed preference data, consistently outperforms PEBBLE. Including original preference

data in the training showed marginal improvements in performance and faster convergence in two out of three environments. However, further validation is required across a more diverse set of environments.

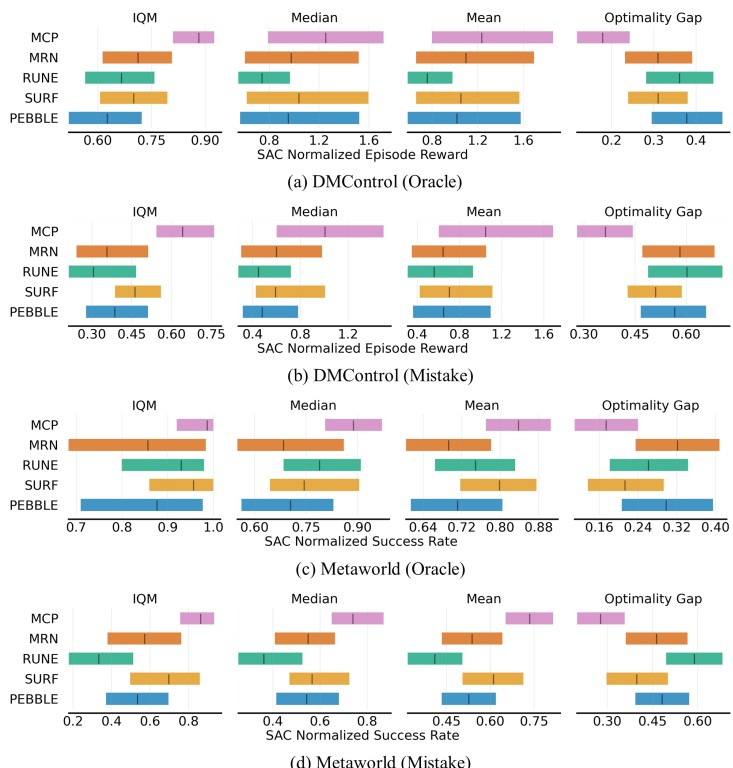

Figure 14: Aggregate metrics for all algorithms across ten independent runs. All metrics, including IQM and OG, are estimated by 95% stratified bootstrap confidence intervals (CIs). The first and second rows show the metrics across the DMControl domains, while others represent the metrics across the Metaworld domains. MCP outperforms other PbRL methods in terms of IQM and OG.

## D.6    RELIABILITY DIAGRAM AND ECE LOSS

MCP, similar to original mixup (Thulasidasan et al., 2019), helps regulate overconfidence. Training with binary preference labels using BCE loss without mixup can lead to overconfidence. This results in a significant divergence between trajectories, with a tendency to assign excessively low rewards to non-preferred trajectories and excessively high rewards to preferred trajectories. MCP moderates this extreme underestimation or overestimation of specific trajectories. To evaluate confidence calibration, we analyzed the reliability diagram and expected calibration error (ECE) (Guo et al., 2017) for both PEBBLE and MCP in two environments (Cheetah and Sweep Into). In the reliability diagram, the gap for each bin between confidence and accuracy is represented by red bar. Among these, the bars with the highest count are highlighted in a darker shade. The large gaps with the lower bar graphs (area below $y = x$) indicates significant overconfidence. A higher ECE suggests that the reward estimator is overconfident. As shown in Figure 16, MCP effectively calibrates the overconfidence of the reward estimator, thereby preventing excessive divergence between trajectories.

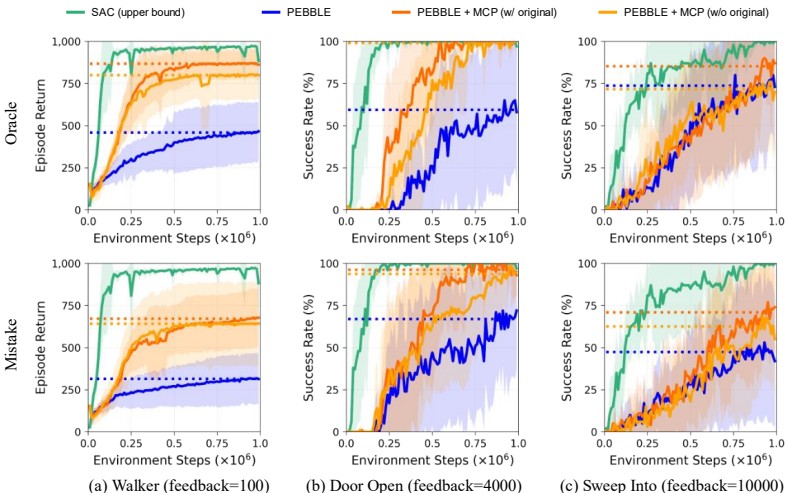

Figure 15: Ablation study on original preference data. The orange line represents MCP's performance when utilizing both original and mixed preferences, while the yellow line depicts MCP's performance solely relying on mixed preferences. Integration with the original preferences improves the MCP's performance marginally regardless of the corruption.

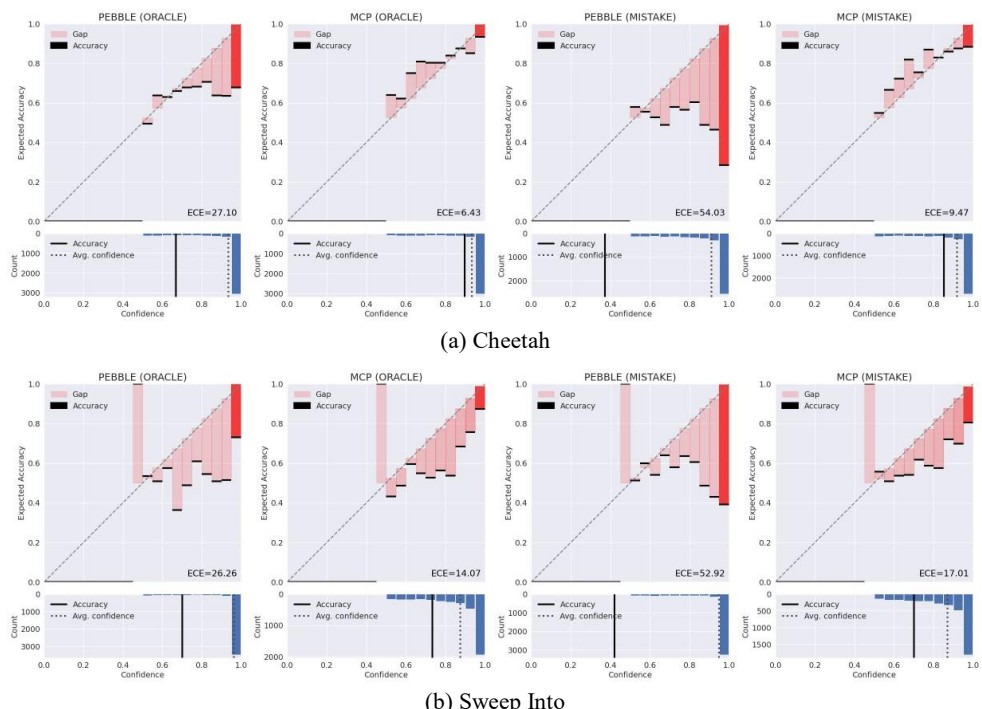

Figure 16: Reliability diagram and ECE Loss for both PEBBLE and MCP. The area of bars below $y = x$ and ECE value indicate the significance of overconfidence.

### D.7 HUMAN ANNOTATION PROTOCOL

To rigorously validate MCP's effectiveness in real-world scenarios, we conducted an analysis with actual human-in-the-loop, beyond the simulated human teacher used in the B-Pref benchmark. We evaluated both MCP and PEBBLE in the Walker environment using actual human annotators (the authors). For a fair comparison, our experimental steps were as follows: First, we ran SAC from

scratch to gather 150 episodes, each containing 1,000 state-action sequences. We then randomly selected 1,000 trajectory segment pairs, each 50 steps long, for annotation. These pairs were distributed among five annotators, with each annotator expressing their preference between two options or voted for equal. For our experiments, 400 preferences were sampled without replacement from the pool of 1,000 annotated preferences for each run. It is important to note that we utilized pre-collected annotations rather than queries generated from ongoing online interactions, ensuring consistency in the feedback process. In each feedback session, the agent received 40 random queries with 400 feedbacks in total. Figure 17 (a) shows the learning curves of MCP compared to PEBBLE under this human-annotation scenario across five independent runs. Figures 17 (b) and (c) display the behaviors of agents trained with human feedback using PEBBLE and MCP, respectively. As depicted in Figure 17 (a), the quantitative difference between PEBBLE and MCP is not significant compared to the results obtained with a simulated human teacher. However, Figure 17 (b) and (c) offer qualitative insights to compare both algorithms. The agent trained with PEBBLE often walks in an unstable posture or crawls forward, while the agent trained with MCP moves forward in a correct posture. MCP demonstrates superior learning compared to PEBBLE, not only in rewarding walking but also in acquiring knowledge about maintaining a stable posture and responding when it falls. The videos for representing depicted trajectories are included in the supplementary.

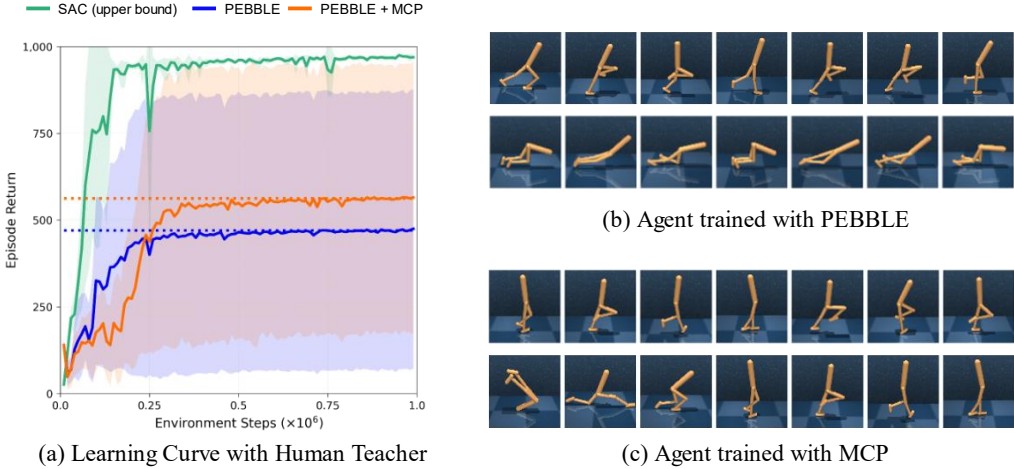

(a) Learning Curve with Human Teacher

(b) Agent trained with PEBBLE

(c) Agent trained with MCP

Figure 17: (a) Learning curves for the Walker task using a human annotation protocol. Mean and standard deviation are estimated from ten independent runs for SAC. For PEBBLE and MCP, these estimates come from five independent runs each. (b) and (c) display the trajectories of agents trained with human feedback, using PEBBLE and MCP respectively.

