# OpenReview forum: "Mixing Corrupted Preferences for Robust and Feedback-Efficient Preference-Based Reinforcement Learning"
_ICLR.cc/2024/Conference — Submitted to ICLR 2024_

### Official Review · Reviewer_2zSd · 2023-10-31

**Soundness:** 2 fair
**Presentation:** 3 good
**Contribution:** 2 fair
**Rating:** 6
**Confidence:** 4

**Summary:**

This paper investigates a critical issue within existing PbRL techniques, particularly focusing on the potential adverse effects stemming from corrupted preferences provided by non-expert human labelers. To address this problem, the authors propose a novel approach called Mixing Corrupted Preferences (MCP). This method involves applying a Mixup technique to augment the data, enhancing the model's robustness against faulty instances and improving feedback efficiency, even when working with limited data. The effectiveness of this method is evaluated against state-of-the-art techniques in a standard PbRL benchmark. The results demonstrate that the proposed MCP method outperforms PEBBLE in terms of both robustness and feedback efficiency, while also effectively complementing SURF. A key contribution of this work lies in introducing the Mixup technique into the PbRL framework, thereby bolstering its robustness and feedback efficiency, particularly when dealing with corrupted preference data.

**Strengths:**

While most recent works concentrate on enhancing the feedback efficiency of PbRL, the authors focus on the challenges stemming from the data quality of collected human preferences, a factor that can profoundly influence the learning process and the performance of the RL agent. This issue is relatively underexplored but holds great significance within the PbRL framework. I appreciate the authors for dedicating their efforts to address this crucial topic.

Furthermore, the idea of the proposed method is elegantly straightforward. It requires only data processing and doesn't necessitate the introduction of complex learning algorithms. This simplicity renders it easily implementable and adaptable to nearly all existing state-of-the-art methods within the conventional PbRL frameworks. In terms of presentation, I feel the paper is well-organized and easy to read.

**Weaknesses:**

In section 3, a strong assumption is made regarding the state and action space. Although the authors briefly mentioned the limitations in the future work section, I believe a more comprehensive discussion of the implications of this assumption is necessary. For the experimental design, I think it could benefit from a more thorough comparison with state-of-the-art PbRL methods, particularly SURF, which also employs data augmentation for PbRL. Additionally, it may be pertinent to consider the paper on RL from diverse humans [1] as a closely related work that approaches the issue of corrupted human preferences as well in a distinct manner. This comparison would be crucial for evaluating the robustness in handling corrupted preferences. However, the authors may not be aware of this work, and they are not the first to address the issue of corrupted preferences. Furthermore, I recommend introducing the Mixup techniques in the preliminary section to enhance the paper's flow. Finally, while I acknowledge the challenges in conducting real human experiments for PbRL in complex control tasks, this work could be further strengthened with user studies conducted with a diverse group of non-expert human labelers.

[1] Xue, W., An, B., Yan, S., & Xu, Z. (2023). Reinforcement Learning from Diverse Human Preferences. arXiv preprint arXiv:2301.11774.

**Questions:**

1. What are the potential negative impact of corrupted preference? What might the newly generated trajectory look like? How do they mitigate the negative impacts? Could you provide some concrete examples for illustration? It is a bit vague to me.

2. Could you briefly explain in which cases the proposed MCP approach may not work effectively?

---

> ### Author Response · Authors · 2023-11-19
> **Responses to Reviewer 2zSd's Comments and a Summary of the Revisions (Part 1/3)**
>
> We sincerely appreciate the reviewer's comments. We are grateful because the reviewer's in-depth reviews improved our manuscript. We have updated our manuscript based on the comments and colored by blue. We respond each comment in detail one by one as below.
> ___
>
> ### **W1. Necessity of convexity assumption**
>
> In considering the assumption of convex states and actions, we concluded that it is not a crucial requirement for our method. The potential lack of real-world convexity in mixed trajectories does not negatively impact network learning. This is similar to the original mixup approach, which interpolates two image-label pairs that might not always be realistic. Despite this, mixup has shown considerable robustness against adversarial examples and incorrect labels, largely due to vicinal risk minimization.
> The primary benefit of mixup is its capacity to generalize and regularize learning process, prioritizing the overall data characteristics over the strict realism of individual augmented samples based on convexity. Moreover, studies involving the augmentation of virtual trajectories have shown significant performance gains without explicitly checking the validity of generated rollouts [1, 2]. Consequently, we have decided to remove the assumptions related to convexity and validity from the description of our method.
>
> References
> [1] Yu, T., Lan, C., Zeng, W., Feng, M., Zhang, Z., & Chen, Z. (2021). Playvirtual: Augmenting cycle-consistent virtual trajectories for reinforcement learning. Advances in Neural Information Processing Systems, 34, 5276-5289.
> [2] Shi, L. X., Lim, J. J., & Lee, Y. (2022). Skill-based model-based reinforcement learning. arXiv preprint arXiv:2207.07560.
> ___
>
> ### **W2. More thorough comparison with state-of-the-art PbRL methods**
>
> Thanks for the insightful comments. We have expanded our experimental comparison to include all environments on SURF, as detailed in Appendix D.4 (Figure 12-14). Furthermore, we have implemented two additional novel preference-based RL methods, RUNE [1] and MRN [2], which were only referenced in the Related Works section of our previous manuscript. We have also updated the tables detailing the hyperparameters for these methods in Appendix C (Table 3-5).
>
> References
> [1] Xinran Liang, Katherine Shu, Kimin Lee, and Pieter Abbeel. Reward uncertainty for exploration in preference-based reinforcement learning. In International Conference on Learning Representations, 2022. URL https://openreview.net/forum?id=OWZVD-l-ZrC.
> [2] Runze Liu, Fengshuo Bai, Yali Du, and Yaodong Yang. Meta-reward-net: Implicitly differentiable reward learning for preference-based reinforcement learning. Advances in Neural Information Processing Systems, 35:22270–22284, 2022.
> ___
>
> ### **W3. Missing related works and comparison**
>
> We appreciate the suggestion to consider the paper on RL from diverse human preferences [1] as a related work addressing corrupted human preferences. We have included the paper in the Related Works section. However, we encountered a challenge in conducting a direct comparison due to the unavailability of their method's code. We have reached out to the authors for access to their code. We are hopeful for a positive response and plan to update our manuscript with additional experiments upon receiving it. Meanwhile, to ensure a comprehensive comparison, we have re-implemented other publicly accessible methods [SURF, RUNE and MRN], as mentioned in our response to W2.
>
> References
> [1] Xue, W., An, B., Yan, S., & Xu, Z. (2023). Reinforcement Learning from Diverse Human Preferences. arXiv preprint arXiv:2301.11774.
> ___
>
> ### **W4. Introducing mixup in the preliminaries**
> Thanks for in-depth review. We have introduced the Mixup technique in an additional preliminary section on Appendix A.

---

> ### Author Response · Authors · 2023-11-19
> **Responses to Reviewer 2zSd's Comments and a Summary of the Revisions (Part 2/3)**
>
> (Continue)
>
> ### **W5. No analysis with actual human-in-the-loop**
>
> We understand the value of including real human annotations in our study to further justify the effectiveness of MCP. However, we faced significant challenges, including the absence of a standardized protocol for human annotation and difficulties in maintaining control over various variables for fair comparison.
> To ensure a rigorous and fair evaluation, we followed the practices common in numerous established preference-based RL studies [1-4]. These studies typically utilize the B-Pref benchmark [5] for evaluations. The B-Pref benchmark is renowned for its effectiveness in simulating human irrationality, making it a robust and controlled environment for methodological comparisons. This benchmark allows us to assess the performance of MCP in a consistent and reliable manner, compensating for the absence of direct human-in-the-loop analysis.
>
> References
> [1] Kimin Lee, Laura M Smith, and Pieter Abbeel. Pebble: Feedback-efficient interactive reinforcement learning via relabeling experience and unsupervised pre-training. In International Conference on Machine Learning, pp. 6152–6163. PMLR, 2021b.
> [2] Jongjin Park, Younggyo Seo, Jinwoo Shin, Honglak Lee, Pieter Abbeel, and Kimin Lee. SURF: Semi-supervised reward learning with data augmentation for feedback-efficient preference-based reinforcement learning. In International Conference on Learning Representations, 2022. URL https://openreview.net/forum?id=TfhfZLQ2EJO.
> [3] Xinran Liang, Katherine Shu, Kimin Lee, and Pieter Abbeel. Reward uncertainty for exploration in preference-based reinforcement learning. In International Conference on Learning Representations, 2022. URL https://openreview.net/forum?id=OWZVD-l-ZrC.
> [4] Runze Liu, Fengshuo Bai, Yali Du, and Yaodong Yang. Meta-reward-net: Implicitly differentiable reward learning for preference-based reinforcement learning. Advances in Neural Information Processing Systems, 35:22270–22284, 2022.
> [5] Kimin Lee, Laura Smith, Anca Dragan, and Pieter Abbeel. B-pref: Benchmarking preference-based reinforcement learning. In Thirty-fifth Conference on Neural Information Processing Systems Datasets and Benchmarks Track (Round 1), 2021a. URL https://openreview.net/ forum?id=ps95-mkHF_.
> ___
>
> ### **Q1-1. Potential negative impact of corrupted preferences and how MCP mitigate the negative impact?**
> The negative impact of corrupted preference data is that it can lead to reward estimator that misinterprets user intentions, causing the agent to act inappropriately. Mixing preference data makes it more difficult to merely memorize the corrupted data, acting as a form of regularization [1]. The second benefit is that MCP improves data efficiency by creating unlimited new data, leading to a smoother reward distribution and enhanced generalization. Lastly, it also mitigates overconfidence problem [2], which can hinder the reward estimator learning. Without mixup, training with binary preference labels using binary cross entropy loss might cause significant reward divergence between two trajectories. This happens due to the tendency to assign overly low rewards to non-preferred trajectories (minimizing the likelihood of selecting non-preferred trajectories) and overly high rewards to preferred trajectories (maximizing the likelihood of selecting preferred trajectories). MCP adjusts for this imbalance in trajectory valuation. To evaluate MCP's confidence calibration, we additionally report the reliability diagram and expected calibration error [3] in Appendix D.6 and Figure 16.
>
> References
> [1] Zhang, H., Cisse, M., Dauphin, Y. N., & Lopez-Paz, D. (2018, February). mixup: Beyond Empirical Risk Minimization. In International Conference on Learning Representations.
> [2] Thulasidasan, S., Chennupati, G., Bilmes, J. A., Bhattacharya, T., & Michalak, S. (2019). On mixup training: Improved calibration and predictive uncertainty for deep neural networks. Advances in Neural Information Processing Systems, 32.
> [3] Guo, C., Pleiss, G., Sun, Y., & Weinberger, K. Q. (2017, July). On calibration of modern neural networks. In International conference on machine learning (pp. 1321-1330). PMLR.
> ___
>
> ### **Q1-2. Illustration and detailed explanation of newly generated trajectory**
>
> To clarify the concept of newly generated trajectories, we have added a visual example in Figure 7 of Appendix B. This illustration shows how two trajectories are mixed. Each trajectory segment consists of state-action sequences of length of $H$. A new trajectory is formed by linearly interpolating these sequences component-wise, based on the mixup ratio $\lambda$. Consequently, each component (state-action) of the mixed trajectory is a linear combination of corresponding components from the two original trajectories at the same indices.

---

> ### Author Response · Authors · 2023-11-19
> **Responses to Reviewer 2zSd's Comments and a Summary of the Revisions (Part 3/3)**
>
> (Continue)
>
> ### **Q2. When might MCP not work effectively?**
>
> As mentioned in the conclusion section, the current implementation of MCP is designed for continuous action spaces. While MCP's reward learning is not inherently limited by the type of action space, experimental validation remains important. We recognize that its effectiveness in scenarios with discrete action spaces remains to be explored. This area will be focus of our future research efforts.

---

> > ### Comment · Reviewer_2zSd · 2023-11-22
> > **Post Rebuttal Comments**
> >
> > Thanks for the authors' responses and the additional experiments, which addressed most of my concerns. Having read both the rebuttal and other reviews, I decided to keep my score because of the concern about the absence of real human experiments and the lack of comparison with a closely relevant prior work. With a meticulous and thorough literature review, the authors should not miss it and could have enough time to make a comparison experimentally.

---

> ### Author Response · Authors · 2023-11-23
> **Author's Response**
>
> Thank you for your feedback and for acknowledging our additional experiments. We understand the reviewer's concerns regarding (1) the absence of real human-in-the-loop experiments and (2) the lack of comparion with the closely relevant work [1].
> In response to the first concern, which other reviewers have also highlighted as important, we have now included human-in-the-loop experiments in our revised manuscript. For more details, please refer to our General Response.
> Regarding the comparison with [1], we agree that it is highly relevant study. However, a direct comparison has been challenging due to the unavailability of public reproduction and verification for [1].
> While we haven't been able to address this concern directly, we believe our work provides a comprehensive analysis, including more baselines, varied environments, a higher number of runs, and additional in-depth analyses.
>
> References
> [1] Xue, W., An, B., Yan, S., & Xu, Z. (2023). Reinforcement Learning from Diverse Human Preferences. arXiv preprint arXiv:2301.11774.

---

> > ### Comment · Reviewer_2zSd · 2023-11-23
> >
> > I appreciate the authors' efforts to include an additional real human experiment. I will increase my score.

---

### Official Review · Reviewer_4pKf · 2023-10-31

**Soundness:** 3 good
**Presentation:** 2 fair
**Contribution:** 2 fair
**Rating:** 6
**Confidence:** 4

**Summary:**

This work introduces Mixed Corrupted Preferences (MCP), a form of mixup for trajectories and preferences for Preference-based Reinforcement Learning (PbRL).

Through experiments with MetaWorld and DeepMind Control Suite (DMC), the paper shows MCP leads to increased feedback robustness, measured by higher final policy performance when training PbRL with highly noisy preferences.

**Strengths:**

* The paper is well-written and easy-to-follow.
* The results are both thorough (multiple tasks with different amounts of feedback) and compelling (with MCP trained with a mistake labeller usually reaching or even beating the performance of PEBBLE trained with an oracle labeller).
* The authors show that MCP is beneficial in combination with other PbRL algorithms that improve sample efficiency.

**Weaknesses:**

* _W1_: No analysis with actual humans-in-the-loop. Given that the main goal of MCP is to increase the robustness of PbRL, verifying whether MCP is indeed beneficial is an important ablation.
* _W2_: The assumption of convex states and actions seems pretty strong, and it is hard to gauge whether it has an actual effect on training.


**[[Post-rebuttal update]]**

Regarding _W1_, the authors added a promising ablation with actual humans-in-the-loop a few hours before the end-of-discussion deadline where MCP qualitatively beats PEBBLE. Unfortunately, I did not have time to thoroughly discuss this addition with the authors and some concerns remain (such as the lack of quantitative differences, or the fact that preferences are collected just from the unsupervised trajectory preferences). It is worth pointing out (as the authors did in the rebuttal) that neither SURF, RUNE, MRN, nor B-Pref (many of which were published on this venue) included actual human-in-the-loop experiments.

For _W2_, the authors removed the requirement for convex state and actions. It remains conceivable that the reward is trained with unfeasible trajectories, but this does not seem to negatively affect the performance of MCP.

**Questions:**

* _Q1_: [Followup from _W2_] Could you provide an analysis of what proportion of states end up being invalid for MetaWorld and DMC? For instance, by randomly mixing up states and actions, and using a simulator to verify whether the interpolated states are valid.
* _Q2_: In equation (8),  what happens if only interpolated trajectories are used?
* _Q3_: How are the initial trajectories that are used for mixup selected? Is it based on uncertainty like in PEBBLE? Does the initial selection affect performance of MCP?
* _Q4_: The authors do not provide any intuition about the role of the beta distribution. Could you explain how you arrived to the values used in the experiments?

**Nitpicks and suggestions (will not affect rating)**

* Please use vector graphics for figures, they are currently very blurry when zoomed in.
* Figure 1's caption needs to be expanded upon. I would also use the extra space could also be used to indicate that the trajectories and the labels are used for reward-learning.
* The description of the plots in Section 4 (page 5), is far away from the actual plots.
* Could you add final performance tables for all the experiments. Sometimes it is hard to see which of the methods does better and by how much.
* For figure 4, could you use a boxplot or a violin plot? It would be interesting to see if there are any spearman correlation outliers in each of the tasks and algorithms.
* In the Related Work section, I suggest changing "discovered" to "investigated", to avoid a debate whether algorithms are discovered or invented.
* In the Related work section, I would incorporate the use of mixup in representation learning and computer vision.
* The limitations of the method are currently stated as future work. I suggest to write them as limitations, since it is clearer to the readers.


**[[Post-rebuttal update]]**

All questions were addressed in the rebuttal. Please refer to the discussion for details.

---

> ### Author Response · Authors · 2023-11-19
> **Responses to Reviewer 4pKf's Comments and a Summary of the Revisions (Part 1/2)**
>
> Thanks for the reviewer's insightful comments on our method. We expect that addressing the concerns above must improve our manuscript. We have updated our manuscript based on the comments and colored by blue. We respond each comment in detail one by one as below.
> ___
>
> ### **W1. No analysis with actual human-in-the-loop**
>
> We understand the value of including real human annotations in our analysis to further justify the efficacy of MCP. However, we encountered challenges, such as the absence of a publicly verified human annotation protocol and difficulties in controlling other variables for a fair comparison.
> To ensure a rigorous and fair evaluation, we followed the practices of numerous established preference-based RL studies [1-4]. These studies typically rely on the B-Pref benchmark [5] for evaluation. The B-Pref benchmark is widely recognized for its ability to simulate human irrationality effectively. We believe that this benchmark provides a suitable and controlled environments for comparing methodologies. It allows us to assess the performance of MCP in a consistent and reliable manner, even without direct human-in-the-loop analysis.
>
> References
> [1] Kimin Lee, Laura M Smith, and Pieter Abbeel. Pebble: Feedback-efficient interactive reinforcement learning via relabeling experience and unsupervised pre-training. In International Conference on Machine Learning, pp. 6152–6163. PMLR, 2021b.
> [2] Jongjin Park, Younggyo Seo, Jinwoo Shin, Honglak Lee, Pieter Abbeel, and Kimin Lee. SURF: Semi-supervised reward learning with data augmentation for feedback-efficient preference-based reinforcement learning. In International Conference on Learning Representations, 2022. URL https://openreview.net/forum?id=TfhfZLQ2EJO.
> [3] Xinran Liang, Katherine Shu, Kimin Lee, and Pieter Abbeel. Reward uncertainty for exploration in preference-based reinforcement learning. In International Conference on Learning Representations, 2022. URL https://openreview.net/forum?id=OWZVD-l-ZrC.
> [4] Runze Liu, Fengshuo Bai, Yali Du, and Yaodong Yang. Meta-reward-net: Implicitly differentiable reward learning for preference-based reinforcement learning. Advances in Neural Information Processing Systems, 35:22270–22284, 2022.
> [5] Kimin Lee, Laura Smith, Anca Dragan, and Pieter Abbeel. B-pref: Benchmarking preference-based reinforcement learning. In Thirty-fifth Conference on Neural Information Processing Systems Datasets and Benchmarks Track (Round 1), 2021a. URL https://openreview.net/ forum?id=ps95-mkHF_.
> ___
>
> ### **W2. Necessity of convexity assumption + Q1. Proportion of invalid state-action pairs**
>
> In considering the assumption of convex states and actions, we concluded that it is not a necessary condition for our method. While the mixed trajectory might not always adhere to real-world convexity, this doesn't negatively impact the network learning. This is similar to the original mixup approach, which interpolates two pairs of image and label that might not represent feasible real-world scenarios. Despite this, mixup has shown considerable robustness against adversarial examples and incorrect labels, largely due to vicinal risk minimization.
> The primary benefit of mixup is its capacity to generalize and regularize learning, focusing on the broader data rather than the precise validity of augmented samples based on convexity. Studies that have augmented virtual trajectories also reported significant performance gains without explicitly checking the validity of generated rollouts [1, 2]. Hence, we have decided to remove the assumptions regarding convexity and validity from the proposed method section to address these concerns.
> Nonetheless, we agree that checking the proportion of valid values is also beneficial. We tried to manually set mixed states in the simulator and input them with mixed actions. Unfortunately, the DMControl environments do not offer this functionality. However, in our exploration with the Metaworld environments, specifically the Button Press scenario, we manually set one million mixed transitions from the replay buffer. We found that all mixed actions were feasible at their respective mixed states. This feasibility arises because all actions in Metaworld environments are unconditionally available, and the mixed values fall within the min-max range of spaces.
>
> References
> [1] Yu, T., Lan, C., Zeng, W., Feng, M., Zhang, Z., & Chen, Z. (2021). Playvirtual: Augmenting cycle-consistent virtual trajectories for reinforcement learning. Advances in Neural Information Processing Systems, 34, 5276-5289.
> [2] Shi, L. X., Lim, J. J., & Lee, Y. (2022). Skill-based model-based reinforcement learning. arXiv preprint arXiv:2207.07560.

---

> ### Author Response · Authors · 2023-11-19
> **Responses to Reviewer 4pKf's Comments and a Summary of the Revisions (Part 2/2)**
>
> (Continue)
>
> ### **Q2. Ablation study with the original trajectories**
>
> Thanks for pointing this out. To address this question, we conducted an ablation study in three environments (Walker, Door Open and Sweep Into) using only mixed preference data. This was done by applying just the second term in Equation (8). Figure 15 depicts the learning curves when using only mixed preference data compared to using both original and mixed preference data. 'Concat (O)' refers the original MCP method using Equation (8), which concatenates original and mixed preferences for learning. On the other hand, 'Concat (X)' indicates MCP trained solely with mixed preference data. Our findings indicate that including original preference data in training leads to marginal performance improvements and faster convergence than training only with mixed preference data. However, it is noteworthy that MCP, when trained solely with mixed preferences, prominently outperforms PEBBLE in two of three environments (Walker and Door Open).
> ___
>
> ### **Q3. How to select the trajectories used for mixup and effect in performance**
>
> We do not have a specific process for selecting trajectories for mixup. As described in Algorithm 1 in Appendix B, within a given preference batch, each preference instance is mixed with another randomly chosen instance from the same batch. This implies that any sampling strategy, such as uncertainty-based sampling, influences only the selection of trajectories for annotation (same as PEBBLE), not the selection of trajectories for mixing within the batch.
> ___
>
> ### **Q4. Role and value selection of Beta distribution**
>
> We employed the Beta distribution for sampling the mixup ratio, following the approach in the original mixup paper [1]. For $\lambda \sim Beta(\alpha,\alpha)$, $\lambda$ tends toward extreme values (either 0 or 1) as $\alpha$ decreases. It leads to a mix that heavily favors one of the original trajectories. Conversely, a higher $\alpha$ results in a more balanced interpolation of the two trajectories. Because the optimal $\alpha$ might differ across domains, we conducted hyperparameter searches for DMControl and Metaworld. The results are described in Table 6 in Appendix D.3.
>
> References
> [1] Hongyi Zhang, Moustapha Cisse, Yann N Dauphin, and David Lopez-Paz. mixup: Beyond empirical risk minimization. In International Conference on Learning Representations, 2018.

---

> > ### Comment · Reviewer_4pKf · 2023-11-21
> > **Response to rebuttal**
> >
> > Thank you for your detailed rebuttal and for the additional experiments you have carried out. In what follows, I will address the specific points we were discussing.
> >
> > -----
> >
> > **W1. No analysis with actual human-in-the-loop.**
> >
> > Though I strongly empathise with the difficulty of verifying with actual humans-in-the-loop, I still think this paper would greatly benefit from such experiments. It is, for instance, conceivable that MCP performs better with actual humans than scripted teachers, we just do not know.
> >
> > I disagree that "_B-Pref benchmark is widely recognized for its ability to simulate human irrationality effectively_" , since B-Pref authors' stated that: (https://openreview.net/forum?id=ps95-mkHF_&noteId=KYlCUFmG0re)
> >
> > > "chances are that none of the biases in our simulated teachers perfectly correlate with real human behavior (rather, human behavior is explained by some combination of some aspects of these with others that we do not yet model in the benchmark)"
> >
> > My understanding is that B-Pref is merely an proxy for actual human behaviour.
> >
> > However, the authors are correct that SURF, RUNE, MRN, and B-Pref did not contain actual human-in-the-loop experiments and still got accepted. I will update my review to indicate this, in case the AC disagrees on the need of actual human-in-the-loop experiments.
> >
> > **W2. Necessity of convexity assumption**
> >
> > Based on the empirical results, I agree that convexity is not necessary for MCP. The updated manuscript reads well on this respect.
> >
> > Likewise, I appreciate that the authors have listed the two main advantages of mixup (robustness and feedback efficiency) in section 3.
> >
> >
> > **Q1. Proportion of invalid state-action pairs**
> >
> > Thank you for trying to check the proportion of valid states in DMC, it is unfortunate that the functionality is not available. That said, I suggest adding the Button Press investigation to the appendix. It might allay the suspicions of readers regarding state feasibility.
> >
> > **Q2. Ablation study with the original trajectories**
> >
> > Thank you for running this experiment. In retrospect, it is not surprising that most of the benefit comes from the interpolated trajectories and preferences (mixup on images has the same behaviour), but it is interesting to see it confirmed here.
> >
> > As a nitpick, the caption of figure 15 does not explain the terms `Concat (O)` nor  `Concat (X)`. Please fix it.
> >
> > **Q3. How to select the trajectories used for mixup and effect in performance**
> >
> > This makes sense, I would still update the manuscript to indicate the type of sampling you are using (presumably, uncertainty-based sampling following PEBBLE).
> >
> > I still suspect that other sampling trajectories may negatively affect MCP, since the within-the-batch variety directly affects the space of interpolated trajectories. However, this is a minor point, since any change in sampling strategy is going to affect foremost the usefulness of the labelled preferences anyway.
> >
> > **Q4. Role and value selection of Beta distribution**
> >
> > Thank you for the explanation and for pointing me to Table 6 in the appendix, I missed this during the original review.
> >
> > In Appendix A, It may be worth further clarifying the behaviour of the distribution as $\alpha$ decreases. Currently, the text only states what happens as $\alpha$ increases, but not as it decreases.
> >
> > ------
> >
> > Based on the fact that most of my questions and one of the weaknesses have been addressed, I will increase my review rating to a 6.
> >
> > That said, I still encourage the authors to incorporate the suggestions from the original review (such as vector graphics and violin plots for figure 4).

---

> > > ### Author Response · Authors · 2023-11-23
> > > **Author's response**
> > >
> > > We are pleased to know that most of the reviewer's concerns have been addressed through our rebuttal. In response to your feedback, we have updated the manuscript, including corrections related to Figure 15 and further clarifying the role of the Beta distribution. We prioritized addressing the common concern about human-in-the-loop analysis, which is detailed in our General Response. We acknowledge the reviewer's suggestions such as vector graphics and violin plot for Figure 4 and will make efforts to incorporate these into our manuscript.  We are committed to refining our work further and will continue making updates to address remaining suggestions by the end.

---

> > > > ### Comment · Reviewer_4pKf · 2023-11-23
> > > >
> > > > Thank you for your response, for updating the manuscript, and for running the additional user-study. I think the paper is in an stronger footing than when first submitted.
> > > >
> > > > I have some remaining concerns about the user study but being so close to the end of the discussion, I will prioritise updating my review for the AC. Unfortunately, at this point, I do not feel comfortable raising my review score further.
> > > >
> > > > I will post the detailed concerns if OpenReview allows it after the end-of-discussion deadline.

---

> > > > > ### Comment · Reviewer_4pKf · 2023-11-23
> > > > > **Remaining user-study concerns**
> > > > >
> > > > > Now that the overall review has been updated, let me expand upon the concerns about the user-study.
> > > > >
> > > > > 1. I originally was worried that the authors are the labellers, which could lead to unconscious bias. In this case, this is likely irrelevant since both PEBBLE and MCP share annotations. If concern 2 was addressed however, this could become an issue. Additionally, the authors would be considered experts, which diminishes the impact of these results (after all the potential of PbRL is to allow non-experts to train complex systems).
> > > > > 2. All the annotations come from the unsupervised trajectory buffer. This is not strictly what PEBBLE does (which selects trajectories from the replay buffer at different times during training based on uncertainty). It would have been preferable to simply replace PEBBLE's scripted teacher with an human teacher, and thus select trajectories to annotate at multiple points during training.
> > > > > 3. The only quantitative result is in a graph. A table with final values and standard deviation would have been preferable as it would have made it easier to put the numbers into context.
> > > > > 4. There is no qualitative comparison with the scripted teacher. These should be included for both MCP and PEBBLE.
> > > > > 5. No information demographic information about the labellers, nor about the amount of time needed to label the trajectories was included.
> > > > >
> > > > > That said the videos attached are very promising, so I advise the authors not to get discouraged and to continue working to address these concerns (regardless of the final outcome of the review).

---

### Official Review · Reviewer_77Am · 2023-11-02

**Soundness:** 2 fair
**Presentation:** 3 good
**Contribution:** 2 fair
**Rating:** 5
**Confidence:** 4

**Summary:**

This paper presents a new preference-based RL method that utilizes mix-up to augment the preference labels. Specifically it samples two pairs of data and mixes up the state and actions with a linear combination and also the label for two segments. The method is mainly evaluated on B-Pref benchmark, showing that the method can improve the performance of baseline methods such as PEBBLE and SURF.

**Strengths:**

- Tackles an interesting and important problem of preference-based RL
- Clear and easy writing
- Extensive experiments

**Weaknesses:**

- The main weakness of the method is it's not clear how exactly this method improves the performance, given that it's not clear what exactly the linear combination of state and action would mean and how using them for learning rewards could improve the robustness. Mixed-up values in each dimension could lie in *valid* state and action space because their value would lie between the min and max value of spaces, but as a whole, it's very unlikely that the state and action would be valid state and actions. For instance, linear combination of two proprioceptive states of 7-dof arm is very likely to lead to physically infeasible joint positions, and there's no guarantee that it'll be meaningfully connected to linear combination of labels. Without thorough investigation and support in this point, it's very difficult to justify the usage of method in many cases.
- Also the main weakness is that a lot of experimental results are not statistically significant, and the improvements over PEBBLE and SURF are very weak. At least the number of seed should be significantly increased and also other metrics from [rliable](https://github.com/google-research/rliable) could be considered.

**Questions:**

Please address my concerns in Weaknesses. The paper is well written and clear so that I don't have a lot of questions. But two main weaknesses are so crucial that I want them to be addressed.

---

> ### Author Response · Authors · 2023-11-19
> **Responses to Reviewer 77Am's Comments and a Summary of the Revisions (Part 1/2)**
>
> We sincerely appreciate the reviewer's meaningful comments on justification and statistical significance about our method. We acknowledge that other readers may also have same concerns and addressing them is crucial. We have updated our manuscript based on the comments and colored by blue. We respond each comment in detail one by one as below.
> ___
>
> ### **Q1. How mixup improves performance and robustness?**
>
> The negative impact of corrupted preference data is that it can lead to reward estimator that misinterprets user intentions, causing the agent to act inappropriately. Mixing preference data makes it more difficult to merely memorize the corrupted data, acting as a form of regularization [1]. The second benefit is that MCP improves data efficiency by creating unlimited new data, leading to a smoother reward distribution and enhanced generalization. Lastly, it also mitigates overconfidence problem [2], which can hinder the reward estimator learning. Without mixup, training with binary preference labels using binary cross entropy loss might cause significant reward divergence between two trajectories. This happens due to the tendency to assign overly low rewards to non-preferred trajectories (minimizing the likelihood of selecting non-preferred trajectories) and overly high rewards to preferred  trajectories (maximizing the likelihood of selecting preferred trajectories). MCP adjusts for this imbalance in trajectory valuation. To evaluate MCP's confidence calibration, we additionally report the reliability diagram and expected calibration error [3] in Appendix D.6 and Figure 16.
>
> References
> [1] Zhang, H., Cisse, M., Dauphin, Y. N., & Lopez-Paz, D. (2018, February). mixup: Beyond Empirical Risk Minimization. In International Conference on Learning Representations.
> [2] Thulasidasan, S., Chennupati, G., Bilmes, J. A., Bhattacharya, T., & Michalak, S. (2019). On mixup training: Improved calibration and predictive uncertainty for deep neural networks. Advances in Neural Information Processing Systems, 32.
> [3] Guo, C., Pleiss, G., Sun, Y., & Weinberger, K. Q. (2017, July). On calibration of modern neural networks. In International conference on machine learning (pp. 1321-1330). PMLR.
> ___
>
> ### **Q2. Invalidity concerns about mixed state-action**
>
> We acknowledge that mixup-augmented data might lead to infeasible state-action pairs in complex real-world problems. However, our findings indicate that these interpolations, even if not entirely feasible in real world, do not hinder the reward estimator learning. This is similar to how the original mixup works by interpolating images and labels, which might not always result in realistic images. Still, the mixup has been effective in enhancing robustness against adversarial examples and incorrect labels, as well as improving data efficiency through vicinal risk minimization.
> The key aspect of mixup lies in its generalization for the network's learning process, as mentioned in the previous question. This prevents the network from merely memorizing the whole data, rather than the strict validity of augmented samples. Furthermore, other studies that create virtual trajectories without verifying their real-world validity have also reported significant performance enhancements [1, 2]. In response to the reviewer's concerns, we have removed the convex and validity assumption in the proposed method section.
>
> References
> [1] Yu, T., Lan, C., Zeng, W., Feng, M., Zhang, Z., & Chen, Z. (2021). Playvirtual: Augmenting cycle-consistent virtual trajectories for reinforcement learning. Advances in Neural Information Processing Systems, 34, 5276-5289.
> [2] Shi, L. X., Lim, J. J., & Lee, Y. (2022). Skill-based model-based reinforcement learning. arXiv preprint arXiv:2207.07560.

---

> ### Author Response · Authors · 2023-11-19
> **Responses to Reviewer 77Am's Comments and a Summary of the Revisions (Part 2/2)**
>
> (Continue)
>
> ### **Q3. Statistical significance**
>
> Thanks for the suggestion. To address the concerns about statistical significance, we have now incorporated metrics from rliable library [1]. We have included these additional metrics in our analysis, including the interquartile mean and optimality gap, which can be found in Appendix D.4 and Figure 14. This approach provides a more comprehensive assessment of our method's performance compared to PEBBLE and SURF.
> In our pursuit of statistically significant comparison, we conducted ten independent runs per each experiment, as is standard in studies like PEBBLE [2] and MRN [3]. This number of runs is double that used by SURF [4] and RUNE [5]. Note that in the field of deep RL, 3-10 runs are typical due to computational constraints. Also, the metrics used in rliable are designed to address such challenge, providing reliable estimates even with a limited number of runs.
>
> References
> [1] Rishabh Agarwal, Max Schwarzer, Pablo Samuel Castro, Aaron C Courville, and Marc Bellemare.
> Deep reinforcement learning at the edge of the statistical precipice. Advances in neural information processing systems, 34:29304–29320, 2021.
> [2] Kimin Lee, Laura M Smith, and Pieter Abbeel. Pebble: Feedback-efficient interactive reinforcement learning via relabeling experience and unsupervised pre-training. In International Conference on Machine Learning, pp. 6152–6163. PMLR, 2021b.
> [3] Runze Liu, Fengshuo Bai, Yali Du, and Yaodong Yang. Meta-reward-net: Implicitly differentiable reward learning for preference-based reinforcement learning. Advances in Neural Information Processing Systems, 35:22270–22284, 2022.
> [4] Jongjin Park, Younggyo Seo, Jinwoo Shin, Honglak Lee, Pieter Abbeel, and Kimin Lee. SURF: Semi-supervised reward learning with data augmentation for feedback-efficient preference-based reinforcement learning. In International Conference on Learning Representations, 2022. URL https://openreview.net/forum?id=TfhfZLQ2EJO.
> [5] Xinran Liang, Katherine Shu, Kimin Lee, and Pieter Abbeel. Reward uncertainty for exploration in preference-based reinforcement learning. In International Conference on Learning Representations, 2022. URL https://openreview.net/forum?id=OWZVD-l-ZrC.

---

> > ### Comment · Reviewer_77Am · 2023-11-21
> >
> > Thanks for the response and updating the draft to reflect the comments. I can see that the proposed method somehow works given its performance, but can't recommend this paper to be accepted yet. I'm willing to update my score to 5, but keeping my current score of 3 to avoid making unnecessary borderline rating -- will update the final score after the internal discussion. This is because the current paper is not providing enough support and analysis why it can be useful and when it can be harmful due to its strong augmentation that can make state and actions not be valid. I think the paper still needs to largely improve in this point by providing more experiments and analysis in diverse domains with different setups (e.g., both state and pixels, different environments where the mixup makes sense and not, ..).  I also agree with other reviewers that the paper need human-in-the-loop experiments. For instance, see [Kim et al., 2023] that showed synthetic labels -- even with some synthetic noises from benchmarks like B-pref -- can give different trends and interpretations when compared to human labels.
> >
> > [Kim et al., 2023] Kim, Changyeon, Jongjin Park, Jinwoo Shin, Honglak Lee, Pieter Abbeel, and Kimin Lee. "Preference transformer: Modeling human preferences using transformers for RL." ICLR 2023

---

> > > ### Author Response · Authors · 2023-11-23
> > > **Author's response**
> > >
> > > Thank you for your detailed evaluation. The reviewer's reference to [1] has helped us recognize that the B-Pref benchmark may not perfectly reflect human irrationality. In response to the common concern about the necessity for human-in-the-loop experiments, we have updated our manuscript to include experimental results with human annotators. For more information, please see our General Response and the revised sections in the manuscript (Appendix D.7).
> > >
> > > References
> > > [1] Kim, C., Park, J., Shin, J., Lee, H., Abbeel, P., & Lee, K. (2022, September). Preference Transformer: Modeling Human Preferences using Transformers for RL. In The Eleventh International Conference on Learning Representations.

---

### Author Response · Authors · 2023-11-23
**General Response - Incorporation of human in the loop analysis**

Dear reviewers and AC,
We sincerely appreicate in-depth and insightful comments., We note that shared concern regarding the absence of real human-in-the-loop analysis in our study. In response, we have conducted additional experiments in the Walker environment using real human annotators. This was based on the human labeling protocol from [1]. The implemental details and the results are described in Appendix D.7 and Figure 17. These results demonstrate the effectiveness of MCP compared to PEBBLE when incorporating human annotators. To maintain transparency and fairness in our evaluation, we have included the human annotator protocol (SAC_records_to_dataset.ipynb) in our supplementary code. The videos of the learned agents with both MCP and PEBBLE are available in the 'video' folder. Furthermore, we have updated supplementray materials to include additional experimental code developed during the rebuttal period, enhancing the reproducibility of our findings.

References
[1] Kim, C., Park, J., Shin, J., Lee, H., Abbeel, P., & Lee, K. (2022, September). Preference Transformer: Modeling Human Preferences using Transformers for RL. In The Eleventh International Conference on Learning Representations.

---

### Meta-Review · Area_Chair_9cZG · 2023-12-06

**Metareview:**

The authors present mixing corrupted preferences (MCP), a new preference-based RL method that utilizes mixup to augment the trajectories and preference labels to improve robustness. Experiments show that MCP improves the performance of baseline methods such as PEBBLE and SURF in terms of both robustness and feedback efficiency.

The reviewers recognized the importance of improving robustness in PbRL and the simplicity of the method (in a good way).  During the rebuttal period, the authors partially resolved reviewers' concerns including additional human-in-the-loop experiments and the reviewers increased scores in response. However, there are still some remaining concerns. The authors are suggested to include more analysis of diverse domains with different setups and analyzing augmented state-actions (suggested by Reviewer 4pKf), and clarification and analysis on human-in-the-loop experiments (suggested by Reviewer 4pKf). By addressing these concerns, this paper could be much stronger in the next submission.

**Justification For Why Not Higher Score:**

The scores were increased to borderline after the rebuttal period, with the clarification and the new human-in-the-loop experiments. However, there are remaining concerns as summarized in meta-review.

**Justification For Why Not Lower Score:**

N/A

---

### Decision · Program_Chairs · 2024-01-16

Reject